# Universally applicable and tunable graph-based coarse-graining for Machine learning force fields

## Abstract

Coarse-grained (CG) force field methods for molecular systems are a crucial tool to simulate large biological macromolecules and are therefore essential for characterisations of biomolecular systems. While state-of-the-art deep learning (DL)-based models for all-atom force fields have improved immensely over recent years, we observe and analyse significant limitations of the currently available approaches for DL-based CG simulations. In this work, we present the first transferable DL-based CG force field approach (i.e., not specific to only one narrowly defined system type) applicable to a wide range of biosystems. To achieve this, our CG algorithm does not rely on hard-coded rules and is tuned to output coarse-grained systems optimised for minimal statistical noise in the ground truth CG forces, which results in significant improvement of model training. Our force field model is also the first CG variant that is based on the MACE architecture and is trained on a custom dataset created by a new approach based on the fragmentation of large biosystems covering protein, RNA and lipid chemistry. We demonstrate that our model can be applied in molecular dynamics simulations to obtain stable and qualitatively accurate trajectories for a variety of systems, while also discussing cases for which we observe limited reliability.

## 1 Introduction

To study large (bio-)molecular frameworks across large time scales, coarse-grained (CG) molecular dynamics (MD) are a powerful tool in computational biochemistry. These types of simulations require specialised force fields that (1) define how a molecular system is transformed into its CG counterpart and (2) are able to output accurate forces acting on the CG beads for a given set of bead positions. CG force fields are significantly more efficient than their all-atom analogues, which becomes crucial as biological macromolecules of interest (e.g., for drug discovery, or in the characterisation of disease development processes (Filipe & Loura, 2022)) can grow to several hundred thousand or even millions of atoms, and MD simulations can require millions of individual force calculations. Similar to classical all-atom force fields (such as AMBER (Salomon-Ferrer et al., 2013) or GROMOS (Scott et al., 1999)), there exist a large variety of classical CG force field approaches that vary widely in their design, i.e., in the CG mappings they construct and in how the inter-bead potentials are defined. For example, a popular choice is Martini (Marrink et al., 2007), which was originally developed for lipid and surfactant systems, but has been extended to a wide range of biosystems (Marrink et al., 2023).

In recent years, machine learning force fields (MLFFs) have been on the rise as a promising alternative to classical force fields as they can deliver accurate force predictions at a highly reduced computational cost (Poltavsky & Tkatchenko, 2021; Wu et al., 2023). MLFF approaches are typically trained on datasets consisting of molecular structures along with their ground truth energies and atomic forces and can therefore achieve a force prediction accuracy close to the reference method they were trained on, e.g., density functional theory (DFT). However, classical force fields typically outperform MLFF models significantly with respect to inference speed. Therefore, it becomes evident that advanced approaches such as ML/MM hybrid methods or coarse-grained MLFF models play a crucial role for pushing the field of ML for molecular simulation towards widespread practical application.

There have been a few notable contributions to the advancement of CG-MLFF models in recent years. They can be divided into two types of approaches. First, approaches where the CG mapping is tabulated exactly for the possible system types, for example, specifying which atoms for each amino acid of a protein get transformed into CG beads (Husic et al., 2020; Charron et al., 2023; Majewski et al., 2023). A popular approach of this kind is $C_\alpha$ coarse-graining, where only $C_\alpha$ atoms of amino acids are kept as CG beads. The two main limitations of these approaches are (i) that the CG mapping strategy is typically not data-driven and hence to some extent arbitrary, and (ii) its limited generalisability to general (bio-)organic chemistry where building a rule-based assignment strategy in this way grows unfeasibly in complexity. The second type of approach is taking a look at more complex coarse-graining schemes, e.g., rooted in machine learning (Chennakesavalu et al., 2023; Wang & Gómez-Bombarelli, 2019; Nasikas et al., 2022). However, while such approaches are in principle able to be more universally applicable across organic systems, they have so far only been applied in non-transferable contexts, i.e., learning a CG force field for a single type of system. One reason for this may be that these approaches are prone to overfitting due to their increased complexity. Furthermore, previous CG-MLFF approaches have been relying on the invariant SchNet architecture for the force field model (Chennakesavalu et al., 2023; Durumeric et al., 2024), which for all-atom force fields has been mostly replaced by other (typically equivariant) architectures in recent studies, for example, MACE (Batatia et al., 2022b), VisNet (Wang et al., 2022), Allegro (Musaelian et al., 2023), or sparse Gaussian Processes (Vandermause et al., 2022). Both Gaussian process-based and Allegro-based force fields have also been applied in the context of CG force fields before (Duschatko et al., 2024; Loose et al., 2023), however, only to build molecule-specific CG-MLFFs with a rigid CG strategy.

We emphasise that in the field of MLFFs, we generally distinguish between system-focused and transferable MLFF approaches. In the former, the model is just trained on a single molecule in different conformations (a single potential energy surface) with the goal to later obtain accurate simulations for this molecule. On the contrary, transferable approaches train on a dataset with a variety of systems with the goal to be applicable to unseen molecules of similar chemistry. As outlined above, the field of CG-MLFFs has largely been focused on the system-focused setup which is common for proof-of-concept studies. One of the reasons for this is that a universally applicable CG mapping is not trivial to define, and as a result we are lacking understanding of whether a truly transferable CG-MLFF approach is even possible. Due to the difficulty of defining a general CG mapping, the few published transferable CG-MLFF approaches are restricted to very specific types of systems, typically proteins where the CG mapping can be fixed for each amino acid in a simple rule-based way.

In this work, we present a new approach for CG-MLFF models trained on a new dataset spanning a vast range of biology-related (organic) chemistry. We build these models based on the MACE architecture which is modified to allow for CG-specific inputs such as continuous multi-dimensional node features. We show that by construction, our model is transferable to unseen structures of similar chemistry and generalisable to larger structures than used for training. The CG mechanism is based on previous work by Webb et al. (2018) which facilitates general applicability to a wide range of structures (in contrast to specialised CG schemes developed for proteins only). It allows us to coarse-grain any (organic) chemical system, which is reflected not only in the fact that our training dataset (see section 2.3) contains lipids, RNA and proteins, but also since the new fragmentation-based dataset generation results in a diverse set of organic fragments to train on (including molecules from the PubChem database). Moreover, we explore an extension to this scheme by adding more physical information in a tunable fashion, and present an ablation study related to this modification. For the force field deep learning model, we apply the state-of-the-art MACE architecture that has recently been demonstrated to be very powerful in the context of all-atom force field foundation models (Batatia et al., 2023; Kovács et al., 2023). However, it is yet to be applied for CG simulations.

## 2 METHODOLOGY

### 2.1 MACHINE LEARNING FORCE FIELD

MACE is a state-of-the-art message passing GNN architecture for interatomic potentials. Its main innovation is the expansion of messages as a hierarchical body order expansion allowing for a decoupling between receptive field size, local geometry modelling and the number of message passing

layers. Hence, even with a small number of model layers, higher order interactions can be captured. Recently, two force field foundation models, Batatia et al. (2023) and Kovács et al. (2023), have been developed based on MACE. For more details on the original MACE model, we refer to the original work, Batatia et al. (2022b) and Batatia et al. (2022a).

We build our implementation based on the JAX implementation of MACE[1] which differs compared to the PyTorch version in some details (see Appendix A.1 for more information). Furthermore, for CG force fields, the original way to embed node features must be adapted. In the MACE full-atom architecture, the atomic number of an atom is mapped to an integer which then corresponds to an index in a trainable parameter matrix. Hence, by construction this is a discrete representation which has its cardinality predefined, i.e., the number of possible atomic species. However, our CG beads have continuous descriptors (i.e., node features), which are constructed as described in section 2.2.3. To allow for this, we apply a multi-layer perceptron (MLP) to the bead descriptor at the first MACE layer. The size of this MLP is a model hyperparameter and its parameters are learned at the same time as the other parameters of MACE during training. We refer the reader to a more detailed description of this matter (and for other small architectural modifications, as well as the model hyperparameters) located in section A.1 in the Appendix.

In the following, we apply MACE as the architecture in all of the trained models. We note that since our paper introduces modifications compared to previous CG-MLFF approaches in multiple ways, we refrain to study the impact of MACE compared to other architectures systematically in this work but rather concentrate on an ablation study regarding the tunability of the CG algorithm. However, in future work, it may be of high interest to assess the influence of the MACE architecture on these results more carefully.

## 2.2 COARSE-GRAINING MODEL

### 2.2.1 REQUIREMENTS

As outlined in section 2.1, an ML force field acts on a set of nodes which can be atoms or beads as long as they have well-defined positions and node features. The process of creating a set of beads with bead positions and bead features from a set of atoms is the coarse-graining process. We present a rule-based approach for this process (based on Webb et al. (2018)), i.e., an algorithm which is pre-defined by us and does not need to be learned based on our reference data. However, our version of this algorithm also contains a small number of hyperparameters which are tuned in a pre-training phase with respect to given quality metrics. Such an approach is discussed further below.

The advantages of such a non-learnable approach is that it can be viewed purely as a pre-processing step both during training of the force field and at inference time (i.e., during an MD simulation). Figure 1 illustrates this set-up for training the coarse-grained ML force field. Note that the coarse-graining cannot adapt to enhance the force field training and any fixed strategy of assigning atoms to beads is arbitrary to some extent. As a result, we do not expect this approach to have the same potential for final accuracy compared to a fully learnable approach, however, its simplicity is expected to be advantageous for model generalisability because the introduced inductive bias acts as a natural regularisation. However, we tune the hyperparameters of our CG model to our training dataset.

For our approach, we define a set of requirements:

1. It must be universally applicable to any chemical system, at a minimum all organic systems, because (1) we want to be able to generalise to a wide range of organic molecules and (2) our training dataset (see section 2.3) does not guarantee any properties of the training structures such as, for instance, if the nucleobases or amino acids would be fragmented always at the same position.

2. It ideally allows for control over the degree of coarse-graining (i.e., the system size reduction factor of fine-grained to coarse-grained representations), which lets us tune this parameter depending on the results during the experimentation phase.

3. It can be designed to include tunable hyperparameters which can be optimised with respect to our dataset in order to alleviate the disadvantages that arise from employing a fixed algorithm for coarse-graining.

---

[1] `https://github.com/ACEsuit/mace-jax` (accessed: 2023-08-11)

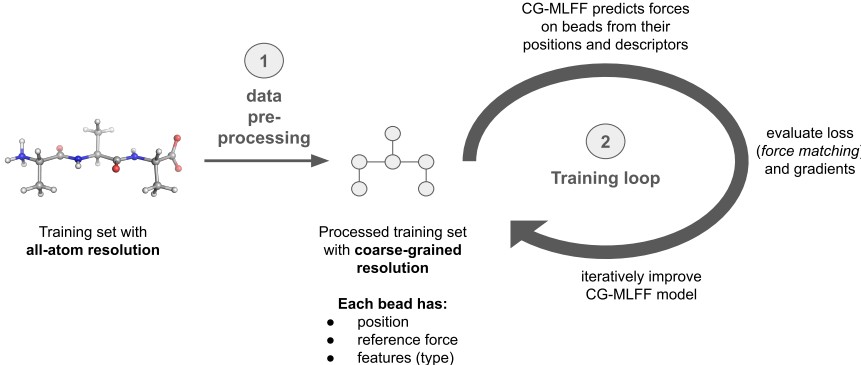

Figure 1: Illustration of the training of the CG-MLFF model. The training is divided in two steps where the coarse-graining is purely a pre-processing step that converts the all-atom training set to a coarse-grained training set.

Furthermore, we emphasise that the coarse-graining process can be split up into three parts conceptually, (1) the atom-to-bead assignment, (2) the weighting of atom positions to form the bead position, and (3) the bead feature (bead type) definition. Our approach handles all of these parts separately. We discuss the theoretical considerations related to designing a coarse-graining model in section 2.2.2, and then describe our designed algorithm in detail in section 2.2.3.

### 2.2.2 PROPERTIES OF CG MAPPING AND FORCE AGGREGATION

A coarse-graining mapping $\mathcal{C}$ transforms the position matrix $R$ of the atoms to the position matrix $\tilde{R}$ of the beads,

$$\tilde{R} = \mathcal{C}(R) \quad . \tag{1}$$

In order to be physically reasonable, $\mathcal{C}$ cannot take any arbitrary form but it must satisfy a few conditions, for instance, that $\mathcal{C}$ must be a linear mapping. For more details, see Appendix A.2.

One challenge of training a coarse-grained ML force field model is that the ground truth forces are only available in all-atom representation. As these need to be compared to the model predictions in coarse-grained representation, we must aggregate the atomic forces to obtain bead forces, i.e., determine a force mapping $\mathcal{C}_F$ that linearly transforms the fine-grained forces $F$ to the coarse-grained forces $\tilde{F}$,

$$\tilde{F} = \mathcal{C}_F(F) = \mathcal{C}_F \cdot F \quad . \tag{2}$$

As first derived by Ciccotti and coworkers (Ciccotti et al., 2005), compatibility with the coarse-grained mapping $\mathcal{C}$ is fulfilled if,

$$\mathcal{C}_F \cdot \mathcal{C}^{\mathrm{T}} = \mathbb{1} \quad , \tag{3}$$

with $\mathbb{1}$ being the identity matrix. Eq. (3) is the only condition to be satisfied in the case that we do not have any constraints on any coordinates, hence, many different $\mathcal{C}_F$ can be chosen given a coarse-grained mapping $\mathcal{C}$. Common approaches are the *pseudoinverse* approach, which solves Eq. (3) straightforwardly and yields,

$$\mathcal{C}_F = \left(\mathcal{C}\mathcal{C}^{\mathrm{T}}\right)^{-1} \cdot \mathcal{C} \quad , \tag{4}$$

or the *uniform* approach, which sums up all atomic forces of all atoms contributing to a bead, hence, setting $(\mathcal{C}_F)_{ij} = 1$ where $\mathcal{C}_{ij} > 0$ and $(\mathcal{C}_F)_{ij} = 0$ otherwise.

However, we adopt the *statistically optimal* approach, as presented by Noé and coworkers and implemented in the open-source Python library `aggforce`. This approach finds an optimal force mapping given the $F$ matrices of $N$ structures sampling a potential energy surface, such that the noise in $\tilde{F}$ is minimal. They demonstrate that this can be achieved by parametrising a force mapping with parameters $\eta$ and optimising these to minimise the overall norm of the forces for the full trajectory,

$$\eta_{\mathrm{opt}} = \arg\min_{\eta} \langle \, || \, \mathcal{C}_F(F; \eta) \, ||_2^2 \, \rangle_F \quad , \tag{5}$$

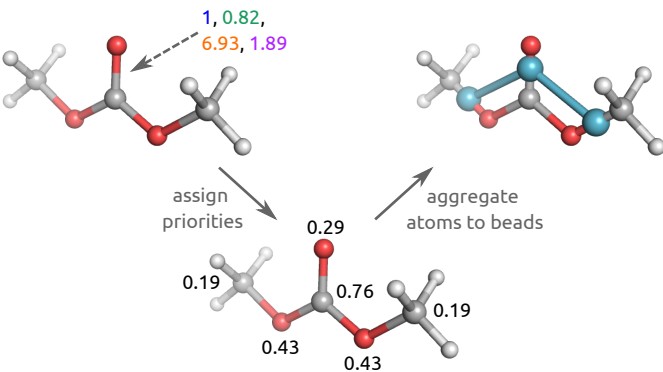

Figure 2: Illustration of the tunable CG model approach. First, we build four chemical graphs differing in edge weighting schemes (weights represented by different colors). Second, the eigendecomposition of the graph Laplacians result in node priorities depicted on the molecule in the centre (only on heavy atoms). Finally, the aggregation algorithm is applied based on these priorities to determine the CG beads and their positions are determined by centre of mass.

with the average running over all sets of all-atom forces $F$ from all configurations (positions $R$) of a given training set molecule. Further details and derivations can be found in Krämer et al. (2023). We will also take advantage of the fact that minimisation of CG force magnitudes results in less noisy training data for our tunable CG approach explained in section 2.2.3.

### 2.2.3 TUNABLE GRAPH-BASED COARSE-GRAINING APPROACH

Our approach is based on the graph-based systematic molecular coarse-graining approach published by Webb et al. (2018). The approach follows a simple two-step process. First, every atom in a molecule is assigned a priority value measuring the importance of an atom for the overall molecular structure. Second, a rule-based algorithm is defined to group atoms together to beads iteratively based on these priority values. The algorithm can be applied iteratively, which leads to even coarser representations, thus making the dimensionality reduction factor controllable. We also explore a modified version of this algorithm which contains tunable hyperparameters for reasons explained in section 2.2.1. The overall approach is illustrated in Figure 2.

As described above, the first step is to assign priorities to each atom. In principle, these could be derived from any algorithm (although an ML-based one would require the second step to be differentiable), however, for simplicity and tunability, we adopt the spectral decomposition method from the original paper and add additional ways to weight the edges of the chemical graph. This method assigns the priority values based on the importance of a node for the overall topology of the chemical graph. In the original paper, a simple binary graph construction based on chemical bonds as edges is employed. This means that atoms that are bonded to many other atoms (either directly or via intermediate bonds) are assigned higher priorities. To obtain these values, a Laplacian matrix $L$ of the molecular graph is constructed,

$$L = D - A \quad , \tag{6}$$

where $D$ is the degree matrix that has the degree, i.e., the number of neighbours, of each node on its diagonal, and $A$ is the adjacency matrix of the graph with the values 1 and 0 as the off-diagonal terms depending on whether two nodes share a chemical bond or not. To determine the node priorities, we perform an eigenvalue decomposition of $L$ and take the absolute values of the coefficients of the first eigenvector (corresponding to largest eigenvalue) as the priority values. The first eigenvector of $L$ contains information about the contribution of each node (i.e., atom) to the overall connectivity in the graph. The more well-connected an atom is within the graph, the higher its contribution. This considers not only the atom's number of neighbours but also its neighbour's neighbours, and more distant connections.

As an extension to this approach, we define multiple graphs describing the chemical system. The graph topology is always based on chemical bonds, however, this time we assign weights $w$ to the

edges depending on physico-chemical properties of the bonds. The Laplacian of these graphs is also computed as defined in Eq. (6) with the adjacency matrix $A$ containing the edge weights ($A_{ij} = w_{ij}$) and the node degree of atom $i$ calculated as the sum of all $w_{ij}$ with $w_{ij}$ as the weight of the $j$-th neighbouring node of $i$. As an addition to the standard binary graph approach, we define three edge weighting schemes:

1. $w_{ij} = 1/R_{ij}$, with $R_{ij}$ as the edge distance.

2. $w_{ij} = \sqrt{Z_i Z_j}$, with $Z_i$ and $Z_j$ as the nuclear charges of atoms $i$ and $j$, respectively.

3. $w_{ij} = 1 + |\chi_i - \chi_j|$, a bond polarity measure, with $\chi_i$ and $\chi_j$ as the Pauling electronegativities of atoms $i$ and $j$, respectively.

The intuition for this choice is that the added properties cover all the straightforwardly computable features of chemical bonds that could be relevant for prioritising the atoms attached to them. We emphasise that using the distance-based features does not imply that we plan to rerun the coarse-graining process at multiple steps during an MD simulation. The CG mapping will remain constant during a simulation and the features will be computed on the initial (possibly optimised) structure.

For the three graph constructions, we compute the normalised Laplacian $L^{\text{norm}}$ instead of the regular one,

$$L_{ij}^{\text{norm}} = L_{ij}/\sqrt{d_i d_j} \quad , \tag{7}$$

with degrees $d_i$ and $d_j$. Otherwise, the influence of the number of neighbours is too strong, which is already covered by the standard binary graph from the original work by Webb et al. (2018).

We compute the spectral decomposition of each of these four Laplacians and perform a linear combination of the resulting node priorities $p$ with coefficients $c$ to obtain the total priority $p_{\text{total},i}$ for each node $i$,

$$p_{\text{total},i} = c_A p_{A,i} + c_B p_{B,i} + c_C p_{C,i} + c_D p_{D,i} \quad . \tag{8}$$

The optimal values for $c_A$, $c_B$, $c_C$, and $c_D$, should be the ones resulting in the most accurate force field models. However, since the second part of the coarse-graining algorithm (described below) is not differentiable, we are limited to optimisation algorithms that do not require gradients and therefore an optimisation target which is fast to evaluate is desirable. Inspired by Eq. (5), we demonstrate in section A.4 in the Appendix that CG models that result in lower average magnitudes of CG forces across the dataset, train better than models with higher average force magnitudes. As a result, we optimise our hyperparameters $c$ with respect to this target, which can be evaluated efficiently for a given CG model without running any force field parameter update steps. Details about this optimisation process are discussed in section A.4 in the Appendix.

Based on the obtained node priority values, we can subsequently build a CG mapping with a rule-based algorithm that iteratively processes atoms and assigns them to beads until all atoms are assigned. Combining this algorithm with an option for weighting the atom contributions yields the final coarse-grained mapping $\mathcal{C}$ described in section 2.2.2. We choose these weights such that the bead position is the centre of mass of the atoms that make up the bead. In Appendix A.3, we explain the bead assignment algorithm in more detail. We also note that we pass only fragments of a pre-determined maximum size to the CG algorithm, which requires a simple fragmentation scheme for larger structures, which we also describe in Appendix A.3.

To finalise the coarse-graining process, bead features need to be assigned. In classical force fields, these would correspond to bead types (typically, a discrete number of types exist). However, we generalise the bead description and assign an $N_{\text{feat}}$-dimensional feature vector to each bead. This vector consists of two parts:

1. The *bead mass* (see section 2.2.2 for its definition).

2. The *element composition* of the bead. This is a 6-dimensional vector containing the number of hydrogen atoms in the bead as the first value, the number of carbons as the second, and continuing for all six elements accounted for in our force field (i.e., H, C, N, O, S, P).

As a result, each bead is characterised by a 7-dimensional feature vector. Additional features were explored during this work, most prominently, features derived (i) from bead-internal distance measures (e.g., mean distance of atoms to bead position) and (ii) from the *Coulomb matrix* descriptor (Rupp et al., 2012) of the bead structure. However, we observed a negative effect of adding these

geometry-dependent features with respect to MD stability of the resulting models indicating that the additional model complexity hampers its generalisability.

## 2.3 DATASET

To train a coarse-grained ML force field, we require a dataset that contains molecular structures with their chemical elements, atomic positions and atomic ground truth forces. We are interested in covering large parts of chemical space relevant for biological systems, in particular, proteins, RNA, and lipid chemistry. Since the coarse-graining process reduces the system sizes significantly, having larger molecular systems up to and beyond 200 atoms in our training set is desirable such that the maximum number of neighbours within the receptive field of the force field models encountered during simulations of large biomolecules is already observed during training. A molecular dataset fulfilling these requirements is currently not available, and therefore, we apply our in-house dataset generation pipeline to generate one.

Our pipeline starts from large molecular structures (e.g., full proteins or RNA) and generates smaller molecular fragments from these, for which the conformation space is sampled (a) via MD, (b) by adding Gaussian noise to MD snapshots, and (c) by stochastic conformer generation. Reference calculations are run on the sampled structures subsequently. In this work, we employ semi-empirical quantum chemical methods as the reference, in particular, the GFN1-xTB method (Bannwarth et al., 2021). Appendix A.5 contains a description of the components of our pipeline in more detail and provides a more in-depth overview of the properties of the resulting dataset. Moreover, we emphasise that our reference dataset does not contain solvated structures. Remarks on the reason behind this choice are made in Appendix A.6.

Our final dataset generated with our pipeline contains 4.9 million structures. We split it randomly into training, validation and test set in a 80:10:10 fashion, keeping all structures (sets of positions) of one fragment in the same subset to prevent data leakage. As mentioned above, further details on the dataset can be found in Appendix A.5.

## 3 RESULTS

We train and evaluate two different models,

- a *standard* model, which generates the CG mapping based on the simple binary graph representation of the molecule (i.e., $c_A = 1$ and $c_B = c_C = c_D = 0$), and

- the *tuned* model, which generates the CG mapping based on the optimised weighting of the four calculated node priorities. The detailed results of how we obtained the optimised weights are located in section A.4 in the Appendix.

The standard model acts as a baseline to assess the improvements achieved by our extension to the original approach by Webb et al. (2018). During training, we monitor multiple metrics of the training and validation set. In Figure 3, we present the learning curves for both models and for three metrics, namely, the loss, the 95th-percentile error of force norms and the Pearson correlation between the predicted and ground truth forces. In particular, we observe that the 95th-percentile error is a useful metric for predicting which model has a high probability to produce stable MD simulations, because occasional inaccurate predictions can self-amplify during a simulation. At the same time, the maximum error on the validation set behaves less predictably between epochs as it depends on the single worst predictions which may be caused by outlier structures in the dataset.

For each metric, the *tuned* model performs better than the *standard* model. Furthermore, we observe that the training curves are significantly smoother for the *tuned* model, hinting at a well-behaved (less noisy) optimisation space. This is especially evident for the Pearson correlation (see right plot in Figure 3). Based on the explanation above, we select our models by their validation set 95th-percentile error. This corresponds to the models after 212 and 101 epochs of training for *standard* and *tuned*, respectively. Contrary to the *standard* model, note that the *tuned* model has not reached its final training epoch of 300. However, since the validation set metrics for this model already surpass the ones for the *standard* model, we decide to run the evaluation on the 101-epoch model.

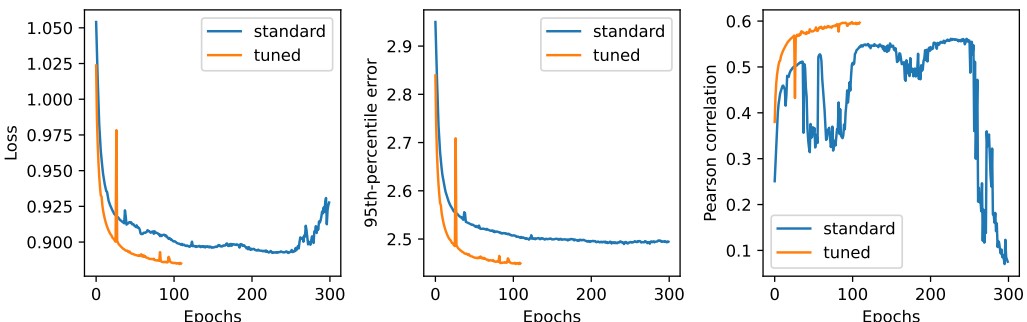

Figure 3: Validation set metrics for the *standard* and *tuned* model during training. We present the loss, 95th-percentile error of the force norms and the Pearson correlation obtained on the validation set. We observe that the *tuned* model outperforms the *standard* model on the validation set.

As the next step, we assess the practical applicability of our models and their generalisability to new systems (in size and composition) by running coarse-grained MD simulations. Our simulations are executed by interfacing the JAX-MD (Schoenholz & Cubuk, 2021) package with the JAX implementation of the models.

First, we assess the stability of MD simulations run with our two ML-based CG force fields across various systems representing protein, RNA, and lipid chemistry. However, we selected test systems which are not part of our training dataset, hence, these are neither fragments generated from larger systems nor part of the PubChem subset. As most of these test systems are directly taken from the RCSB Protein Data Bank, we refer to these by their PDB ID. For this stability test, we run 100 ps of NVT Langevin dynamics with a timestep of 5 fs at 300 K. Unstable MD simulations in which the system topology breaks apart is a common issue for ML force fields, especially early in their training process (Brunken et al., 2024). Apart from a visual inspection of the trajectory, this behaviour can be detected easily by observing the kinetic energy of the system, or the temperature. The latter is kept roughly constant in NVT simulations if the trajectory is stable. In Table 4 of Appendix A.7, we present the maximum temperatures encountered during the 100 ps simulations of all our test systems for the *standard* and the *tuned* model. We exclude the first five picoseconds of simulation time to make sure the system has time to thermally equilibrate. We note that in all-atom FF contexts, additional stability measures are commonly applied (Hoogeboom et al., 2022), however, their extension to the CG context is not straightforward. Furthermore, we provide visualisations of four of the coarse-grained test systems overlaid with their all-atom counterparts in the Appendix A.7 in Figure 10. Table 4 demonstrates that we obtain across stable simulations across most test systems (RNA, lipids, proteins) for small systems of less than one hundred atoms and for larger systems of multiple hundred atoms. However, some systems also break apart during the simulation for both ML models (e.g., PDB ID: 1BQF) or just for the *tuned* model (e.g., PDB ID: 1P79). Further visual inspection of the trajectories also show bond stretching between beads for some examples, most notably for the weak phosphorus−oxygen bonds in the phosphate groups of RNA structures. We provide images of MD snapshots that demonstrate this phenomenon in Appendix A.7 in Figure 9. Moreover, with respect to MD stability, we do not observe the superiority of the *tuned* model over the *standard* one but instead we document a weak opposite effect. We hypothesise that a reason for the lack of improvement observed in MD stability with the *tuned* model may be that the main metric we optimise for during training and CG model tuning is related to the average of forces, however, MD stability is mainly related to individual inaccurate predictions along a long-running MD trajectory. This is a well-known issue not only for CG force fields, but also for all-atom ones, and may be a plausible reason for our observed stability issue. For stable systems, we also run 10 ns simulations to assess (a) the model stability for longer simulation times and (b) the computation time obtained for the simulations in our set up. The results are reported in Appendix A.7 in Figures 11 and 12.

Second, we take one of our test systems that proved to be stable for both of our models and assess the qualitative physical accuracy of the stable simulation further. For this, we compare our trajectories to the ones of (1) the Martini CG force field (see Appendix A.8 for details), and (2) the GFN-FF all-

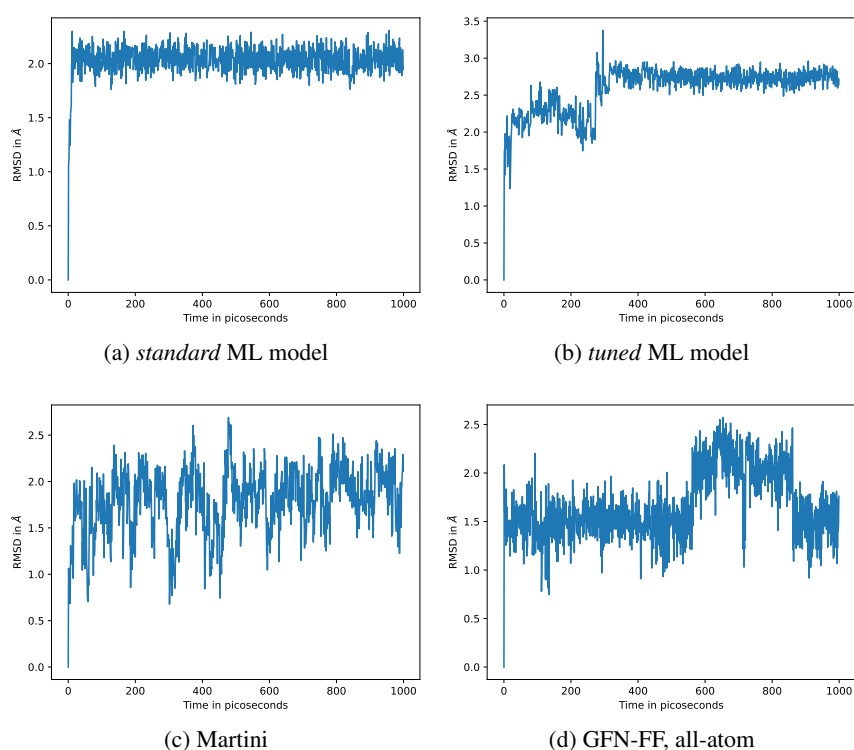

(a) *standard* ML model

(b) *tuned* ML model

(c) Martini

(d) GFN-FF, all-atom

Figure 4: Comparison of RMSD along 1 ns trajectories for four CG force fields (two ML, one CG reference, and one all-atom reference) with respect to the identical set of initial positions.

atom force field (Spicher & Grimme, 2020). Of course, the comparability between these trajectories is limited as the simulated systems do not share the same dimensionality, however, our main goal is to assess whether our trajectories are qualitatively reasonable beyond pure stability, e.g., with respect to sampling similar conformations. As a test system, we selected the IPP lactotripeptide (taken from the complex with PDB ID 8QFX). For comparison between models, we plot the root-mean-square deviation (RMSD) between the initial positions (which are the same for each model) and the positions along the trajectory (see Figure 4). By inspecting the MD trajectories visually, we observe that in each simulation, the initial structure is quickly transformed into a slightly more folded (i.e., compact) conformation. Depending on the FF model, this conformation has an RMSD of 1.5 and 2 Å compared to the initial structure. Furthermore, a second conformation is observed within the first nanosecond of simulation. Visual inspection of the trajectory hints at the fact that the side chains of the two more distant amino acids move further away from each other in this conformation. It is visited between 600 and 900 ps in the all-atom reference simulation. The only other simulation that visits a similar conformation in this simulation is the *tuned* ML model, however, since the Martini model only uses 6 beads to represent the system, the two conformations may be less clearly distinguishable. To compare between our two ML models, we observe that the *standard* model exhibits very monotonous dynamics while the *tuned* model's dynamics matches the stochasticity of the two reference simulations to a larger extent, and hence appears more physically reasonable because it produces a potential energy landscape where the energy barriers between conformations are less extreme than the ones obtained with the *standard* model. This emphasises our key finding that tuning the CG model is beneficial for downstream force field accuracy. Lastly, we observe larger amplitudes for the reference dynamics compared to our ML models, however, it is uncertain whether this must be attributed to limitations in the model or is caused by varying simulation time steps and dimensionality of the system representation.

For longer simulations of 10 ns, we switch to time-lagged independent component analysis (TICA) plots for visualisation. These plots are commonly used to present protein dynamics with a 2D representation. In Figure 5, we compare the *tuned* ML model with the Martini reference. In both 10 ns

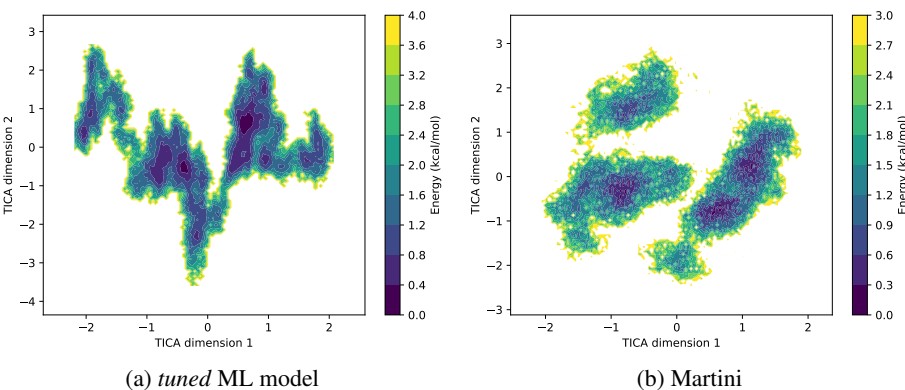

(a) *tuned* ML model        (b) Martini

Figure 5: TICA plots for *tuned* ML model compared to Martini for a 10 ns trajectory. For the TICA plot of the *standard* ML model, see Figure 8 in Appendix A.7.

simulations, we observe three main configurations and a similar energy range between regions that are visited the most and least often, which underlines the key result that the tuned ML model reproduces the dynamics in a qualitatively correct manner. However, we also observe some differences, most prominently that the high energy regions between local-minimum configurations are sampled at a slightly higher rate with the ML model (see the connections between regions in the TICA plot (a) in Figure 5). For the *standard* ML model, the TICA plot matches the reference to a significantly smaller extent (see Figure 8 in Appendix A.7). As mentioned for the RMSD analysis above, the limited comparability between different CG models hampers a conclusive quantitative comparison, however, we plan to tackle this challenge as a subsequent step in future work by generating *in silico* results that can be compared to an experimental reference.

## 4 CONCLUSION AND OUTLOOK

In this work, we introduced a new transferable CG-MLFF model which is not restricted for application to specific types of systems (e.g., only non-fragmented peptide sequences) but can be universally applied to most molecular biosystems. We emphasise that demonstrating proper transferability across different types of systems (i.e., proteins, RNA, lipids, small molecules) is the core contribution of this work as this has not been demonstrated in the CG-MLFF context before. Moreover, we introduced a simple tuning strategy for the graph-based coarse-graining scheme by Webb et al. (2018). Our pipeline comprises the dataset generation, CG model, MLFF model training, and the application of the trained model to new systems in MD simulations which we observed to be stable for most of our test systems. Furthermore, we proved qualitative correctness of our approach, by comparison with the well-established Martini force field and an GFN-FF all-atom reference, for a test system that exhibited a high degree of MD stability. While observing that tuning the CG model significantly improves the MLFF model training, our results on how this advantage translates to MD simulations has not yet been fully conclusive.

In conclusion, we can report two key findings, (1) it is possible to develop a transferable MLFF for coarse-grained systems despite the vastness of (bio-)chemical space and the information loss introduced by the CG mapping, and (2) even a simple and straightforward tunability scheme with four optimisable parameters is able to significantly improve the resulting MLFF model training. As a consequence, we believe that making the CG model even more flexible can further improve the downstream MLFF accuracy leading to increased MD stability. Following this work, the subsequent step is to conduct a more extensive evaluation study of these models across more test systems against Martini and all-atom FFs, and extend these findings to the application of large, research-relevant biosystems. To obtain valuable quantitative results for such systems, we anticipate that (a) treatment of solvation, (b) a dataset with more accurate reference forces (at DFT level), and (c) further fine-tuning of the model to experimental data will be necessary.

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

## A   APPENDIX

### A.1   DETAILS OF THE APPLIED MACE MODEL

In the main text, we noted that the JAX implementation we base our model on contains differences compared to the PyTorch version. These are (1) the implementation of the symmetric contraction (unchanged in our version), (2) the use of irreps-aware linear layers instead of a tensor product for skip connection layers (unchanged in our version), and (3) the output of the symmetric contraction

and subsequent irreps-aware linear layer being set to ignore all irreps beyond $l = 0$ for all layers while it should do so only for the last layer (solved in our version), and some other minor changes.

Furthermore, as also stated in the main text, we apply an MLP to the bead descriptor at the first MACE layer to account for continuous multi-dimensional bead types required for CG modelling. The original MACE uses the following for a 2-body message for the first layer,

$$A_{i,kl1m1}^{(1)} = \sum_{j \in \mathcal{N}(i)} R_{kl1}^1(r_{ij}) Y_{l1}^{m_1}(\hat{\mathbf{r}}_{ji}) W_{k\tilde{z_j}}^1 \quad , \tag{9}$$

where $R_{kl1}^1(r_{ij})$ is a learnable projection of the distance $r_{ij}$ onto a Bessel basis, $Y_{l1}^{m_1}(\hat{\mathbf{r}}_{ji})$ is the output of a spherical harmonic component $l_1 m_1$ at the vector $\hat{\mathbf{r}}_{ji}$, and $\tilde{z}_j$ is the mapping into the integer index of the atomic number $z$ for atom $j$ of a learnable matrix $W^1$ of size *embedding size* $k \times$ *number of atomic species*. In contrast, for the CG (i.e., continuous) case, this message is as follows,

$$A_{i,kl1m1}^{(1)} = \sum_{j \in \mathcal{N}(i)} R_{kl1}^1(r_{ij}) Y_{l1}^{m_1}(\hat{\mathbf{r}}_{ji}) \, \mathrm{MLP}(\mathcal{D}_j)_k \quad , \tag{10}$$

where $\mathcal{D}_j$ is the set of scalar initial bead descriptors for bead $j$, comparable to the use of an atomic number.

A similar modification is applied for multi-body correlation where a similar discretisation is present. The original multi-body message is,

$$B_{i,\eta_\nu kLM}^{(t)} = \sum_{\mathbf{lm}} \mathcal{C}_{\eta_\nu \mathbf{lm}}^{LM} \prod_{\xi=1}^{\nu} \sum_{\tilde{k}} w_{k\tilde{k}l_\xi}^{(t)} A_{i,\tilde{k}l_\xi m_\xi}^{(t)}, \ \mathbf{lm} = (l_1 m_1, ..., l_\nu m_\nu) \tag{11}$$

$$m_{i,kLM}^{(t)} = \sum_{\nu} \sum_{\eta_\nu} \mathrm{W}_{\tilde{z}_i kL,\eta_\nu}^{(t)} B_{i,\eta_\nu kLM}^{(t)} \quad , \tag{12}$$

where $\nu$ is the correlation order, $\mathcal{C}_{\eta_\nu \mathbf{lm}}^{LM}$ are the generalised Clebsch−Gordan coefficients, $\eta_\nu$ index all the possible couplings of $l$ through the tensor product elevated to the power $\nu$ that will lead to the equivariance $L$. The message aggregation into the final message that will be associated to node $i$ is a linear learnable function of the form $B_{i,\eta_\nu kLM}^{(t)}$ via the learnable weight matrix $\mathrm{W}_{\tilde{z}_i kL,\eta_\nu}^{(t)}$ which has one if its dimensions dedicated to encoding the atomic number $z_i$ using the $\tilde{z}_i$ mapping $z_i$ to $\mathrm{W}^{(t)}$ entries. We rewrite the multi-body message as,

$$m_{i,kLM}^{(t)} = \sum_{\nu} \sum_{\eta_\nu} \sigma(\mathrm{MLP}^{(t,\eta_\nu)}(\mathcal{D}_i)_{kL}) \mathrm{W}_{kL,\eta_\nu}^{(t)} B_{i,\eta_\nu kLM}^{(t)} \quad , \tag{13}$$

where again $\mathcal{D}_i$ is the set of scalar initial bead descriptors for bead $i$ and the sigmoid function $\sigma$ rescales the MLP for improved training stability.

Moreover, it was not possible to evaluate the average energy of a bead type in contrast to atom types. Hence, in contrast to the original MACE, our MACE layers do not output an energy deviation from an average or predefined node type energy, but rather the absolute energy value for a bead. Whereas the original MACE weights every component of the 2-body message by a single value representing the average number of neighbours of an atom, we replace it by a per-message component (i.e., a per-edge weight),

$$A_{i,kl1m1}^{(1)} = \sum_{j \in \mathcal{N}(i)} \frac{1}{\sqrt{d_i d_j}} R_{kl1}^1(r_{ij}) Y_{l1}^{m_1}(\hat{\mathbf{r}}_{ji}) \, \mathrm{MLP}(\mathcal{D}_j)_k \quad , \tag{14}$$

where $d_i$ is the degree of node $i$. This approach allows us to train a model more agnostic to the bead densities across various molecules.

Finally, we document the hyperparameters of our MACE model in Table 1 and the configuration of our AMSGrad optimiser for model training in Table 2.

## A.2 Constraints on the CG mapping

As mentioned in the main text, the CG mapping $\mathcal{C}$ cannot take any arbitrary form to be physically reasonable, which in this context refers to the coarse-grained system being thermodynamically

Table 1: List of MACE hyperparameters employed in this work. Note that $m$ stands for mulitplicity and $\sum_m$ is the sum of multiplicities for irreps in a message.

| hyperparameter | value |
|---|---|
| hidden irreps | $128x0e + 128x1o$ |
| output irreps | $2x0e$ |
| symmetric tensor product basis | True |
| off diagonal | True |
| maximum L | 3 |
| irreps considered for interaction | spherical harmonic ($O(3)$) |
| correlation (number of MACE layers) | 2 |
| readout MLP irreps | $16x0e + 8x1o$ |
| radial basis | Bessel functions |
| number of basis functions | 8 |
| radial envelope | soft envelope |
| gate | SiLU |
| MLP for initial bead descriptors embedding | [64,128], swish activation |
| MLP for L type at correlation order $\xi$ | [128,128 $\cdot m$], swish activation |
| MLP radial basis projection | [64,64,64,$\sum_m$], SiLU activation |

Table 2: List of configuration parameters for the employed AMSGrad optimiser during MACE model training.

| parameter | value |
|---|---|
| batch size | 500 |
| learning rate | $10^{-2}$ |
| weight decay | $5 \cdot 10^{-4}$ |
| EMA decay | 0.99 |
| gradient clipping factor | 10 |

consistent with the fine-grained representation. This means that the potential energy surface of the coarse-grained system is consistent with the corresponding one for the all-atom system. For example, two conformations of a molecular system (i.e., two local minima on the potential energy surface) must exhibit the same energy difference in both representations such that the coarse-grained MD samples the two conformations in the same ratio as the fine-grained MD does.

More rigorously, this means that an effective potential $\tilde{U}(\tilde{R})$ may be defined in a thermodynamically consistent manner as the conditional free energy of the coarse-grained configuration, i.e.,

$$\tilde{U}(\tilde{R}) = -k_B T \ln \int_{R \in \mathcal{C}^{-1}(\tilde{R})} \exp\left(-\frac{U(R)}{k_B T}\right) \frac{\mathrm{d}R}{\mathrm{d}\tilde{R}} \quad , \tag{15}$$

with $k_B$ as the Boltzmann constant and $T$ as the temperature. Note that removing the constraint $\mathcal{C}(R) = \tilde{R}$ on the integrand of Eq. (15) would yield the free energy of the fine-grained system (a scalar, equal to $\mathbb{E}[U] - TS$ at equilibrium). Therefore, Eq. (15) can be interpreted as the free energy of the fine-grained system, subject to the condition that coarse-grained positions are $\tilde{R}$. With an identical coarse-graining ($\mathcal{C} : R \mapsto R$), Eq. (15) yields $\tilde{U} = U$, thus preserving the energy function. Further derivations yield a few simple conditions that need to be satisfied for $\mathcal{C}$ to be thermodynamically consistent. We report these conditions below, however, for more details on the mathematical background, we refer to Noid et al. (2008) as well as Wang & Gómez-Bombarelli (2019).

First, the mapping $\mathcal{C}$ must be linear, i.e., the transformation of coordinates is a matrix multiplication,

$$\tilde{R} = \mathcal{C} \cdot R \quad . \tag{16}$$

Furthermore, the elements $\mathcal{C}_{ij}$ of $\mathcal{C}$ must satisfy the following conditions.

- Each row must be normalised, i.e., the weights of the atom contributions to a bead sum to one, and weights below zero are not allowed.

$$\sum_j \mathcal{C}_{ij} = 1 \quad \text{and} \quad \mathcal{C}_{ij} \geq 0 \tag{17}$$

- Each atom contributes to at most one bead.

With this mapping, the $i$-th bead mass $M_i$ can be obtained from the atom masses $m_j$ by the following equation,

$$M_i = \left( \sum_{j=1}^{N_{\mathrm{at}}} \frac{\mathcal{C}_{ij}^2}{m_j} \right)^{-1} \quad , \tag{18}$$

where $N_{\mathrm{at}}$ is the number of atoms.

### A.3   BEAD ASSIGNMENT ALGORITHM AND PREPROCESSING

In this section, we describe the algorithm that assigns atoms to beads depending on their respective priorities determined in the previous step (see the main text). This algorithm is taken from the original work of Webb and coworkers (Webb et al., 2018).

1. Remove all hydrogen atoms from the structure. They are added to the beads which contain their bond neighbours later. In principle, this step is optional and the hydrogen atoms can also be treated as all other atoms.

2. Start with the atom with the lowest priority. If multiple atoms have the same priority, they need to be processed at the same time. Find the neighbouring atom that is (i) not assigned to a bead yet, (ii) has a higher priority than the currently processed atom, and (iii) has the priority that is closest to the one of the currently processed atom. Group these two atoms together, which creates a new bead. If multiple neighbours have the same priority, it can also happen that the resulting bead will contain more than two atoms.

3. Iterate through all atoms from lowest to highest priority and process them as described above. Note that if all of the neighbours of a given atom are already assigned to a bead, then this atom will make up a new bead by itself.

4. After one full pass through all atoms, the created beads are the result of the first coarse-graining level. If needed, one can repeat this procedure $N$ times to obtain a system coarse-grained to level $N$. Before starting a new iteration, we need to assign new bonds between beads to construct a graph, which can be achieved by drawing chemical bonds between two beads if any of the atoms that make up the beads were bonded before.

Up to this point, the described algorithm determines which atoms contribute to which beads, however, the weights of these contributions are still undetermined. Multiple schemes to determine these weights are possible, for example, using the centre of mass of the atoms of a bead as the bead position, the centre of positions, the centre of positions excluding hydrogens, and many others. We only apply the center-of-mass method in this work. Combining the algorithm described above with this option for weighting the atom contributions yields the final coarse-grained mapping $\mathcal{C}$ described in section 2.2.2 of the main text.

Furthermore, we note that we implement a simple molecular fragmentation scheme to pass only fragments of maximum size $N_{\mathrm{frag}}^{\max}$ to the CG algorithm. We perform this step because (1) the Laplacian matrix size grows quadratically with fragment size potentially resulting in memory issues for very large systems, and (2) the ranking of eigenvector coefficients becomes numerically unstable in the case of a large number of atoms. Our fragmentation scheme first separates the unconnected subgraphs of the system (i.e., the individual molecules) and then iteratively cuts them in half until

Table 3: Pearson correlation between various metrics computed on training runs of various CG models (grid search of priority weights). The metrics under investigation are the training set losses after $N$ epochs $L_N$ ($N = 5, 10, 15, 20$) as well as the mean magnitude of CG forces $\langle \| \tilde{F} \| \rangle$ across the dataset.

|  | $L_5$ | $L_{10}$ | $L_{15}$ | $L_{20}$ | $\langle \| \tilde{F} \| \rangle$ |
|---|---|---|---|---|---|
| $L_5$ | 1 | 0.79 | 0.82 | 0.87 | 0.70 |
| $L_{10}$ | 0.79 | 1 | 0.99 | 0.97 | 0.94 |
| $L_{15}$ | 0.82 | 0.99 | 1 | 0.99 | 0.94 |
| $L_{20}$ | 0.87 | 0.97 | 0.99 | 1 | 0.92 |
| $\langle \| \tilde{F} \| \rangle$ | 0.70 | 0.94 | 0.94 | 0.92 | 1 |

a maximum fragment size of $N_{\text{frag}}^{\text{max}}$ is reached. The resulting fragments are then passed to the CG model separately. For each system type in our dataset, we define possible cuts, e.g., peptide bonds for proteins, $O-C$ bonds next to phosphate groups in RNA, and simple $C-C$ single bonds for lipids. We apply a value of $N_{\text{frag}}^{\text{max}} = 80$ in our experiments.

### A.4    TUNING THE CG MODEL

As described in section 2.2.3, our objective is to optimise the weights for the individual node priorities in Eq. (8) such that the resulting model trains well on our dataset.

Inspired by Eq. (5), the mean magnitude of the CG forces is potentially a very convenient measure for the quality of a CG model (i.e., a set of priority weights) as it is directly related to the amount of statistical noise in the ground truth values learned by the MACE force field model. To validate this assumption in practice, we train MACE models on a small subset of the data that is coarse-grained by various CG models that have been determined by a grid search over the priority weights (following values for each weight: 0, 0.1, 1, and 10). For the training set, we selected 100 fragments from the original dataset at random, each with all of their associated configurations (sets of positions). We record the training set loss after 5, 10, 15, and 20 epochs, as well as the mean magnitude of the CG forces (which can be calculated without the need for any training steps). We emphasise that these are the ground truth aggregated CG forces, and not the ones predicted by our model (which at this stage has not been trained yet).

We present the Pearson correlation between grid points for each pair of metrics. Spearman rank correlation coefficients were also calculated and exhibit identical qualitative trends. As expected, our results show that the performance of our CG-MLFF models converges with increasing number epochs, for example, the performances after 5 and 10 epochs are only correlated with a coefficient of 0.79, while performances after 15 and 20 epochs are highly correlated at 0.99. Based on these results, 10 epochs of training seem to be sufficient to identify the optimal CG models in an optimisation procedure. However, we also observe that the mean magnitude of CG forces is strongly correlated with the losses after 15 and 20 epochs, and therefore, this metric is very powerful for our CG model optimisation because we can score a given CG model efficiently without having to perform any CG-MLFF model training. The full set of results can be found in Table 3.

For CG model tuning, we apply the Differential Evolution algorithm as implemented in the `SciPy` package (Virtanen et al., 2020). The bounds for the four priority weights were set to 0 and 1, the relative tolerance of convergence to 0.01, and the dataset size was increased to 200 fragments (with all their configurations) as the computation of the mean force magnitudes, which acts as our minimisation target, is highly efficient. We obtain the following optimised hyperparameters for our CG model: $c_A = 0.73$ for the binary graph, $c_B = 0.08$ for the $1/R_{ij}$ weighted graph, $c_C = 0.18$ for the $\sqrt{Z_i Z_j}$ weighted graph, and $c_D = 0.08$ for the $1 + |\chi_i - \chi_j|$ weighted graph.

## A.5 DATASET GENERATION

In this section of the Appendix, we provide more details on our in-house dataset generation pipeline, which can be divided into the following three steps.

First, we generate our fragments directly from large systems (proteins, RNA, lipids nanostructures) by an automated fragmentation procedure employing the fragmentation procedure of the `Swoose` C++ package (Brunken et al., 2021) implemented as part of its automated force field parametrisation functionality (Brunken & Reiher, 2020). Because the molecular fragments should represent a similar region of chemical space as the local substructures in the large biosystems of interest for MD applications, we generate our fragments directly from such systems. This fragmentation procedure cuts out spherical fragments from a larger structure with a radius $r_{cut}$ and then follows the cut bonds recursively up to a bond that is an allowed bond to cut with a well-defined saturation scheme. `Swoose` implements this procedure for protein systems and it also works for most organic systems, however, we further extend it by adding another possible bond cut, which is cutting $O-P$ bonds and replacing them with $O-H$. This allows us to generate more diverse fragments for RNA molecules. Furthermore, `Swoose` generates one fragment around each atom in the system, which would result in too many fragments per system for our purpose. Hence, we keep only a subset of fragments sampled in a way to maximise the distances between the atoms around which the fragments were generated. Moreover, we increase the diversity of covered chemical space by including a selection of small molecules from the PubChem subset of the SPICE dataset as additional fragments (Eastman et al., 2023).

Second, based on the fragments and PubChem molecules as initial structures, we sample the conformation space, i.e., the potential energy surface, by three distinct methods, described in the following.

1. We run GFN-FF (Spicher & Grimme, 2020) MD simulations at 500 K and extract snapshots in equidistant intervals. We select the GFN-FF method as a well-established universal classical force field with fixed bond topology, such that we avoid chemical reactions during the sampling process at the high simulation temperature as we previously observed in Phys-Net's solvated fragments dataset (Unke & Meuwly, 2019). The most significant advantage of the MD sampling method is that is produces physically reasonably configurations that are generated in a well-defined process. However, its disadvantages are that for short-to-medium length MD runs, the extracted snapshots can be autocorrelated to some extent and only a local part of the full conformation space can be sampled. Hence, we add methods 2 and 3 to our sampling approach.

2. For every $N$-th snapshot extracted from the MD simulation, we generate $N$ random sets of positions that we obtain by applying small displacements to the original positions. These displacements are sampled from a Gaussian distribution. This `mdgauss` method, extends the `md` method by adding a set of uncorrelated configurations. The number of structures obtained from this sampling method is by construction the same as the number of structures obtained from the MD run.

3. By applying the `OpenBabel` software (O'Boyle et al., 2011), we add conformers that are generated by a stochastic approach and cover the conformer space more broadly. We specifically apply this software as it allows conformer generation even for systems that contain more than one molecule, which occurs regularly in our generated fragments. The number of conformers generated can, in principle, be freely chosen, however, for simplicity, we generate the same amount as the number of structures sampled from MD runs. Note that for small fragments or PubChem molecules it may be possible that `OpenBabel` is not able to generate that many distinct conformers. In such cases, we generate the maximum possible number of conformers.

For an example fragment, we present the sampling of geometries visually in Figure 6.

Lastly, one requires reference forces for the generated structures. These can be calculated at any level of quantum chemical approximation. For the sake of this proof-of-concept study, we employ semi-empirical quantum chemical methods as the reference, in particular, the GFN1-xTB method implemented in the `xtb` software (Bannwarth et al., 2021). Semi-empirical methods are known to exhibit higher accuracy compared to classical force fields as they already contain quantum mechanical information for our ML force field model to learn. Moreover, the higher efficiency of

semi-empirical methods compared to density functional theory allows us to generate a very large dataset, which has been shown to be necessary for CG force field training, as the coarse-graining process adds noise to the data (Krämer et al., 2023).

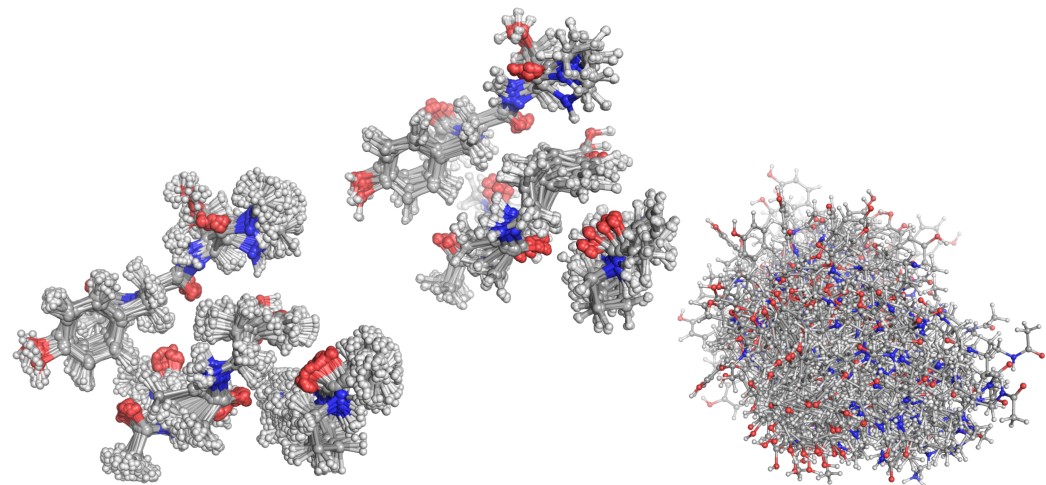

Figure 6: Overlay of all sampled geometries for one peptide example fragment generated from PDB ID *1OGA*. The geometries are separated by sampling method, namely `md` sampling (on the left), `mdgauss` sampling (in the center), and `OpenBabel` sampling (on the right). This illustration demonstrates the strengths and weaknesses of each of the sampling methods, most notably that the MD-based methods sample the geometry more locally while the stochastic conformer generation allows the sample the potential energy surface in a more diverse manner.

For this work, we generate a dataset focused on fragments of protein, RNA, and lipid systems as these are the systems of interest for this project due to their high relevance for computational studies of biological systems. The original large structures used for fragmentation and the resulting dataset will be provided alongside this work.

We sample 400 structures for each of the sampling methods `md` and `mdgauss`, and 100 additional structures with `OpenBabel` for each molecule/fragment. As stated above, the number of sampled structures with the `OpenBabel` method may be smaller for small fragments. The fragmentation was done with an initial radius of 7 Å. The resulting size distribution of all structures in the dataset is depicted in Figure 7. We add a sanity check of the structures generated by the pipeline which filters out physically unreasonable structures. This sanity check verifies that all hydrogen atoms in the structures have exactly one neighbouring atom and that none of the atom pairs are too close to each other. By this filter, we exclude approximately 16% of data points before training. The resulting dataset contains 4,910,710 structures with 7170 unique fragments or molecules.

## A.6   REMARK ON SOLVATION

We stress that solvation plays an important role in all biological systems and typically is explicitly or at least implicitly treated in classical MD simulations. Therefore, it may come as a surprise that we deliberately refrain from adding solvation to our training dataset. We decide to do so for two main reasons,

1. because including solvation adds significant additional complexity to our research problem, e.g.,

   • the coarse-graining of the solvent is not straightforward with our presented approaches and requires an extension of our algorithm, and

   • the training structures as well as the large evaluation structures would grow in size significantly by including a proper number of solvent molecules,

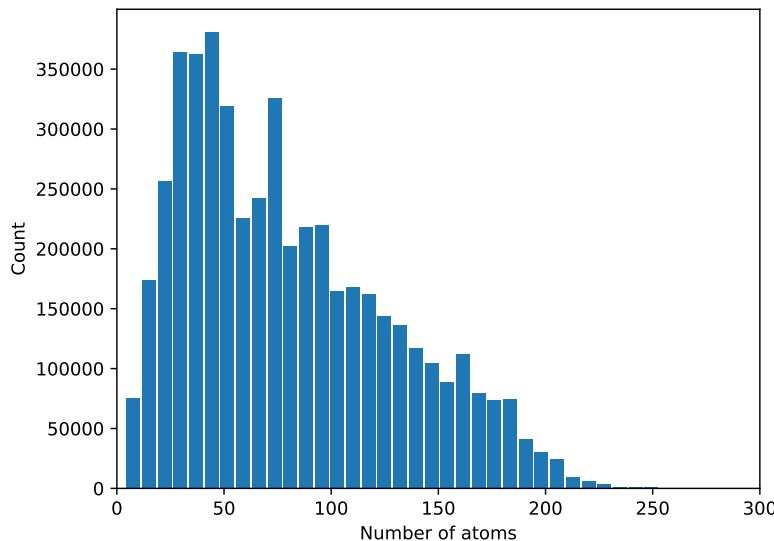

Figure 7: Distribution of system sizes for all 4.9 million structures in our biosystems dataset. The largest fragment consists of 297 atoms.

2. and because we are able to add the treatment of solvation to our models in a subsequent step without compromising the advances made in this work. To achieve this, we require an extension of the dataset with structures that include solvation and retrain our final models on these data after extending the CG algorithm to include a reasonable coarse-graining strategy for solvent molecules.

However, we recognise that without this extension, the quantitative correctness of our models acting on non-solvated structures is limited and for some structures this may possibly hamper qualitative correctness as well. Especially for RNA systems, we can expect that missing solvation may amplify stability problems as RNA molecules are negatively charged due to the phosphate groups.

## A.7 MD SIMULATION ASSESSMENT

This section contains additional material gathered during the assessment of the MD simulations. Table 4 list our test systems along with their sizes and maximum temperatures encountered during 100 ps simulations (to assess MD stability). Figure 10 visualises four of these systems in their all-atom and CG representations. Moreover, we present the TICA plot obtained with the *standard* model for the IPP lactotripeptide test system in Figure 8. In Figures 11 and 12, we present our results for 10 ns simulations for the standard and tuned models, respectively. This includes the computational scaling and stability analysis for the systems that were observed to be stable throughout the 100 ps simulations. These figures demonstrate (i) that many systems remain stable until the end of the 10 ns simulation or at least for a large proportion of it, and (ii) that the computational scaling of the method is approximately linear with system size. However, note that for some systems that break apart early in the simulation, the model's inference efficiency can be increased artificially because the model is dealing with less tightly connected graphs. Lastly, we visualise the observed bond stretches in RNA systems in Figure 9.

Table 4: Overview of 100 ps MD simulations at 300 K run with the *standard* and *tuned* CG force field models including the size of each system (atoms and beads) and the maximum temperature $T$ encountered in the trajectory. Values significantly higher than 300 K signal moments of high kinetic energy release, showing that a given simulation was unstable and the system most likely (at least partially) broke apart.

| Test system | $N^{\text{atoms}}$ | $N^{\text{beads}}_{\text{standard}}$ | $N^{\text{beads}}_{\text{tuned}}$ | $T^{\text{max}}_{\text{standard}}$ | $T^{\text{max}}_{\text{tuned}}$ |
|---|---|---|---|---|---|
| lactotripeptide IPP[2] | 50 | 12 | 13 | 515 | 491 |
| 1AKG | 211 | 58 | 62 | 424 | 396 |
| 8-CHL[3] | 592 | 120 | 136 | 358 | 352 |
| 1CEK | 103 | 28 | 30 | $1.03 \cdot 10^3$ | $1.23 \cdot 10^5$ |
| 472D | 524 | 172 | 194 | 345 | 346 |
| 1BQF | 371 | 94 | 105 | $1.35 \cdot 10^7$ | $6.08 \cdot 10^5$ |
| 5KGZ | 634 | 151 | 177 | 352 | 350 |
| 1P79 | 168 | 57 | 65 | 389 | $2.23 \cdot 10^4$ |
| 1KUW | 139 | 34 | 41 | 447 | 775 |
| 2Z75[4] | 3102 | 1127 | 1191 | 310 | $1.94 \cdot 10^{11}$ |
| RNA fragm. from dataset[5] | 157 | 51 | 55 | 390 | 399 |
| lipid fragm. from dataset[6] | 126 | 25 | 27 | 482 | 437 |

---

[2]Structure taken from complex with PDB ID 8QFX.

[3]A non-bonded cluster of eight randomly placed cholesterol molecules

[4]Full RNA system 2Z75 was fragmented as part of training set generation.

[5]Fragmented from RNA system with PDB ID 5V3I.

[6]Fragmented from large nanoparticle system made up of DOTAP and CHEMS lipids.

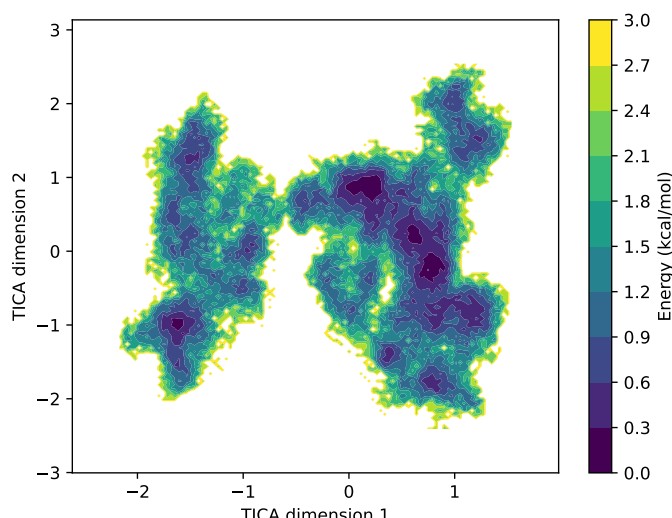

Figure 8: TICA plot for *standard* ML model for a 10 ns trajectory. For comparison with the Martini reference, see Figure 5.

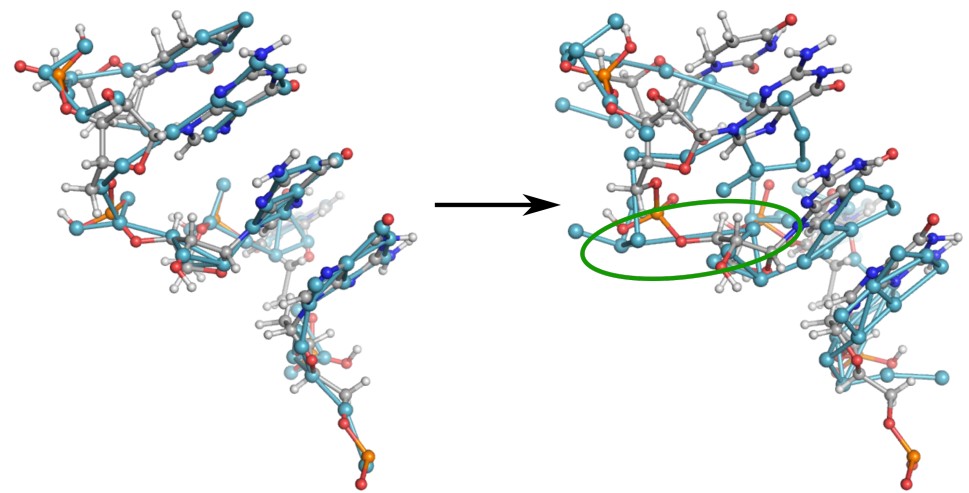

Figure 9: Visualisation of bond stretches (next to phosphate groups) observed during MD simulations of the RNA systems, demonstrated with the example of the system with PDB ID 1P79. On the left, the initial structure is depicted, while on the right, we present the system after 250 ps of simulation. The stretched bond is marked by a green ellipse.

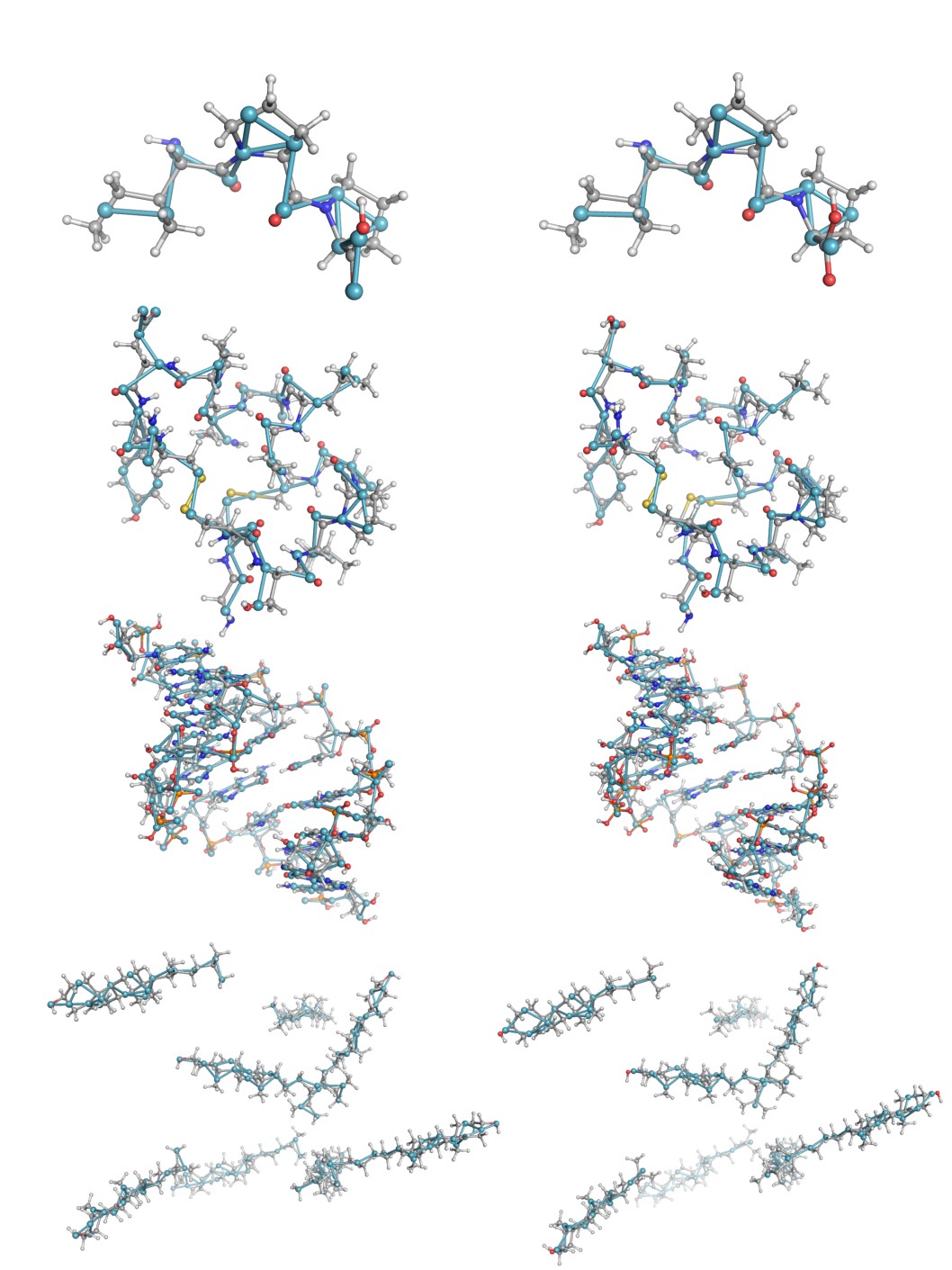

Figure 10: Overlays of all-atom and CG representations of four test systems employed during this work (see Table 4). The CG representations obtained with the *tuned* model are shown on the left, while the ones obtained with the *standard* model are depicted on the right. The systems are from top to bottom: lactotripeptide IPP, protein with PDB ID 1AKG, RNA with PDB ID 472D, 8 randomly-placed cholesterol molecules.

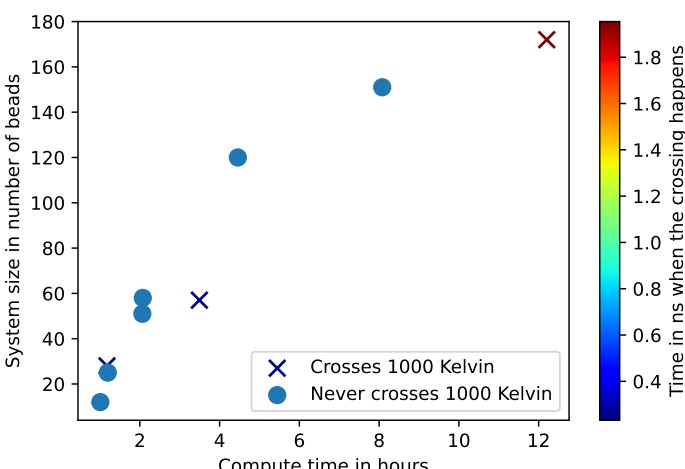

Figure 11: Computational scaling and MD stability analysis for 10 ns simulations with the *standard* CG-MLFF model. MD stability is assessed via the maximum temperature encountered in the trajectory as explained in the main text.

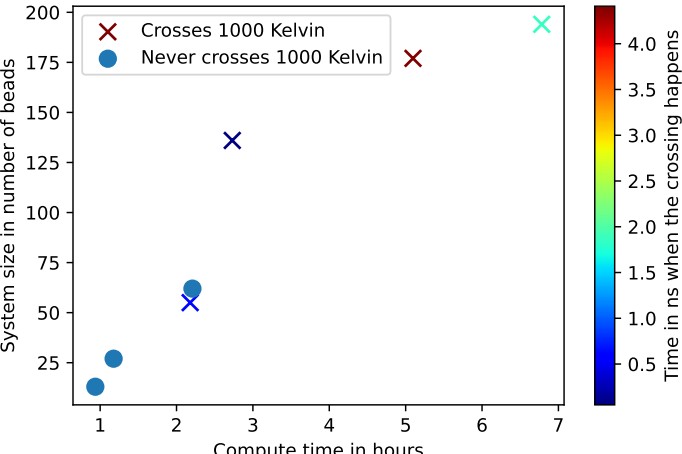

Figure 12: Computational scaling and MD stability analysis for 10 ns simulations with the *tuned* CG-MLFF model. MD stability is assessed via the maximum temperature encountered in the trajectory as explained in the main text.

## A.8 MARTINI SIMULATIONS

In this work, we apply the Martini force field as a CG reference model (Marrink et al., 2007; Souza et al., 2021). Martini uses a simplified representation of molecules, where four heavy atoms are merged into a single coarse-grained (CG) bead (possible mapping is 4-to-1 or 3-to-1) (Marrink et al., 2004). The basic assumption while parametrising of the CG model: carefully parameterised properties of the individual beads are transferable to the whole molecule. The Martini CG scheme is used to study a wide range of biological systems: lipid bilayers, transmembrane proteins, RNA/DNA, membrane fusion, and protein-protein interactions. This approach is very useful for studying biological processes that occur on longer timescales ($\approx 10 - 100\,\mu$s). The most common and first application of the Martini CG approach is, for example, self-assembly of nanoparticles or vesicles.

The systems in this work were modelled using a coarse-grained approach within the canonical (NVT) ensemble. Gromacs 2023 (Bekker et al., 1993) coupled with the MARTINI.v2 force field was employed. The timestep was 10 fs and the temperature was maintained at 300 K, controlled by the Nosé–Hoover thermostat. The system was first equilibrated in the NPT ensemble with a timestep of 1 fs at a temperature of 300 K until the system volume stabilised, which typically required approximately 100 ps. During equilibration, we employed the velocity-rescaling thermostat and the Berendsen barostat. Furthermore, water beads were completely removed from the simulation box, and the Dry Martini force field was applied (Arnarez et al., 2015). Dry Martini is a version of Martini where water beads are removed from the simulation. Interactions between beads are adapted by tweaking the interaction levels and tuning bonded parameters to reproduce the properties observed with the standard Martini force field.

