# OpenReview forum: "Universally Applicable And Tunable Graph-Based Coarse-Graining For Machine Learning Force Fields"
_ICLR.cc/2025/Conference — Submitted to ICLR 2025_

### Official Review · Reviewer_ZwoN · 2024-10-18

**Soundness:** 2
**Presentation:** 1
**Contribution:** 2
**Rating:** 3
**Confidence:** 3

**Summary:**

The paper introduces an extension to a coarse-graining method for force fields: The previous work coarse grains the molecule by assigning priorities to atoms and then iteratively summarizes nodes based on these priorities. The priorities are assigned based on the first eigenvector of the Laplacian. The current paper proposes to use not just the Laplacian matrix but instead constructs weighted Laplacians with the same edges but weighted by three different features, resulting in four separate Laplacian matrices. The priorities are then calculated as a linear combination of the four first eigenvectors of the four matrices. The weights in the linear combination are found by optimizing an objective that is easy to calculate. The authors claim that this extension leads to a universally applicable method due to the higher degree of flexibility compared to the baseline. They support this by showing that their method leads to better training and qualitatively correct behaviours on their tested systems. To test their method, the authors also introduce a new dataset with large molecules designed to better test coarse-graining methods over a variety of systems.

**Strengths:**

The scalability of force fields is an essential open problem and still hinders their applicability in many domains despite big increases in computing power over recent years. Coarse-graining is an underexplored but promising avenue to contribute to faster force fields. I am not an expert on coarse-graining, but it looks like there is a lack of publicly available datasets with a large variety of systems that are simultaneously big enough to assess coarse-graining methods. Therefore, I think the authors are tackling a significant open problem with their work. The dataset alone would be a good contribution if there were some better defined automatically measurable metrics. I would encourage the authors to flesh out this part of their work.
The proposed method for coarse-graining is based on accepted work and is a sensible, although straightforward, extension that shows promising results. I also appreciate that the authors were honest and discussed the limitations of their method's somewhat reduced stability, but I am missing more discussion on this topic.

**Weaknesses:**

I think the work generally has potential but is lacking polish:

The authors use the term "fragments" in the main text (abstract and line 150 and 151) and only explain what fragments are way later in the text, which is confusing because it can be confused with the "beads"

In line 124 its stated that "the simplicity is expected to be advantageous for model generalizability because the introduced inductive bias acts as a natural regularisation". However, I don't see where the inductive bias comes in. The eigenvector of the Laplacian of a molecular graph isn't a physically meaningful concept, as far as I know.

Equation 5 is central to the paper, and an intuitive explanation would be great if possible.

What is the intuition for the three introduced weights? Why do we expect these to be meaningful, and couldn't other ones also be helpful? Since this is the central contribution of the work, a discussion of these aspects is necessary.

The 1/R_ij features change over time and, therefore, can shuffle the priorities, leading to different coarse-grainings for different time steps. Wouldn't this induce unsmooth behaviour in the simulation? If I understand this correctly, since the original work uses only chemical bonds, this wouldn't be the case for the original work unless there is bond breaking.

In line 296: Is the element composition really the number of elements? Could this lead to cases where a lot of elements of the same type are coarse-grained, leading to a large input number, something that is known to make neural networks unstable?

Line 340: how can you just use 100 fragments? Dont we need the entire system to correctly predict energies/forces?

Line 342: what loss do you use?

Line 372: person correlation between what?

Figure 3: This looks promising, but why don't we (also) look at validation/test curves?

Line 402: Why did the tuned model not reach the final training epoch?

Line 428: It is great that the authors were honest and mentioned the reduced stability, but possible reasons for this should be discussed, even if the issue has not yet been solved

Figure 4: I am not sure what these plots are trying to tell me, especially because they are not comparable between the different models, as mentioned in the text. It would be better to replace them with more helpful metrics. For example, if you can visually make out different conformers in the ground truth simulation, why not coarse-grain them with the different methods and then calculate the distance between the model's trajectories and these coarse-grained ground truth conformers? This would make it much easier to see if the different models found the same conformers. It doesn't have to be precisely this metric, but anything more informative would be great.

**Questions:**

Why are only first eigenvectors used?

Does any of this consider bond-breaking?

---

> ### Author Response · Authors · 2024-11-22
>
> We thank you for your detailed review of our manuscript and the well-explained summary of our work.
>
> **Weaknesses**
>
> **I think...**  The extension of Webb et al. is only one of our contributions. Our main contribution is a significant step towards a truly generalizable CG-MLFF model, i.e., one that is transferable across different systems of diverse (organic) chemistry. Such an approach has not been presented before. We present a complete pipeline for: (1) the generation of a chemically diverse dataset (2) a versatile and tunable CG mapping, and (3) the application of a state-of-the-art MLFF architecture (MACE).
>
> **The authors...** We will add a statement of explanation close to line 150 to avoid the confusion.
>
> **In line...** The 1st eigenvector of the graph Laplacian contains information about the contribution of each node (i.e., atom) to the overall connectivity. The more connected an atom, the higher its contribution. Considering not only the atom’s neighbors but also its neighbor’s neighbors, and so on. Assigning high priorities to well-connected atoms, ensures the overall bond topology is maintained when coarse-graining.
>
> **Equation...** For an intuitive explanation, first, it is important to understand that there exist many valid thermodynamically-consistent force mappings given a CG mapping. By coarse-graining, there is loss of information about the all-atom forces, however, some mappings add more statistical noise than others. Kramer et al. derives that a minimal CG force magnitude correlates with the minimal noise in CG forces.
>
> **What....** A more detailed discussion of this will benefit the paper. The heuristics include readily computable features of the chemical bonds, which could be relevant for prioritizing the atoms attached to them; Bond distance; bond polarity, and essentially “bond mass”. We are open to further suggestions for future versions of the model.
>
> **The 1/...** Modifications at each step of simulation would lead to unsmooth behavior. Our coarse-graining remains constant, and based on the bond distances of the minimum energy structure at the start of simulation (roughly equal to the expected value of the bond distance during the simulation). During training, input structures are not optimized, however, many configurations of the same bond are sampled in the training set (with the same expected value), which should regularize the model to not overfit one specific distance for a given bond (if in reality, it fluctuates during a simulation).
>
> **In line...** It is the number of elements, the proportions of each element has also been tested, leading to worse results. We understand the concern, however here the CG strategy will group between 1 and 10 atoms to form a bead. If moving to more coarse-graining, the models may be revisited.
>
> **Line...** This subset is only used for tuning the CG parameters. 100 fragments -> 3000 data points, each fragment comes with all conformations. Training on “entire systems”, though not as many compared to the full force field training later in the pipeline. Taking more fragments for the CG tuning could improve model, but this has not been investigated rigorously in this work.
>
> **Line 34...** For the force field training, we use the same loss as regular training of the ML force field, the same as in the original MACE code, i.e., the MAE of the forces.
>
> **Line...** It is the Pearson correlation between the predicted and ground truth forces.
>
> **Fig...** The presented curve does not show the data on the training set but rather for the validation set. This is mentioned in the caption. We will rephrase to “validation set metrics for the standard and tuned model during training.”.
>
> **Line...** This was due to time constraints. However, the model has finished training, and the results are qualitatively similar, hence, the training had already almost converged at that point.
>
> **Line 428...** The tuned model may slightly overfit to the training data leading to less stable MD simulations for unseen systems. Little improvement in MD stability under tuning may also be due to the shared optimization of the average forces metric during training and CG model tuning. Ultimately MD stability is related to inaccurate individual predictions along a trajectory. This is a well-known issue not only for CG force fields, but also for all-atom ones, and the principal reason.
>
> **Figure...** In comparison between models, Figure 4 demonstrates more physically reasonable simulations from the tuned compared to standard model. The energy barriers between different conformations are high for the standard model, it is stuck in one configuration. In all other simulations, especially the GFN-FF reference, we observe multiple conformations being visited, which is also observed for the tuned model, demonstrating more accurate recovery of relative energies of conformations.
>
> We will address all these elaborated points in the updated manuscript.

---

> > ### Comment · Reviewer_ZwoN · 2024-11-26
> >
> > I thank the authors for answering my questions. I dont think a revised paper has been uploaded so far, so I cannot check any improvements made to the paper.

---

### Official Review · Reviewer_ezmK · 2024-10-30

**Soundness:** 2
**Presentation:** 2
**Contribution:** 1
**Rating:** 3
**Confidence:** 4

**Summary:**

The paper presents a deep learning (DL)-based approach for coarse-grained (CG) molecular dynamics (MD) simulations, targeting large biomolecules like proteins, RNA, and lipids. It builds on a graph-based coarse-graining method, adding tunability to the priority values assigned to atoms, which group them into CG beads based on structural importance. The model aims to create more transferable CG simulations across a wide range of biosystems and improve training stability by reducing noise in force predictions. This work uses the MACE architecture for the force field model and trains it on a custom dataset derived from fragmented large biomolecules.

**Strengths:**

1. **Efficient Coarse-Graining with Tunable Parameters**: The authors' introduction of tunable priority values in the graph-based coarse-graining process allows for optimization based on the dataset. This tunability provides flexibility in CG mapping, allowing it to balance accuracy and computational efficiency effectively.

2. **Application of MACE Architecture**: Leveraging the advanced MACE architecture, a message-passing neural network, allows the model to capture complex interatomic potentials with high accuracy. The architecture’s flexibility and power make it well-suited to the demands of coarse-grained molecular dynamics, potentially enabling more stable and accurate simulations.

3. **Extensive Custom Dataset**: The authors created a robust dataset of 4.9 million biomolecular fragments, supporting model training across varied chemical structures. This comprehensive dataset could facilitate the model’s ability to generalize to new systems, enhancing its reliability in MD simulations.

**Weaknesses:**

###

1. **Unclear Parameter Tuning for Coarse-Grained (CG) Models**: The paper lacks clarity on whether the CG parameters were tuned using the MLFF-predicted forces or the DFT-aggregated forces at the CG level. If the tuning was based on MLFF-predicted forces, it would be valuable to consider tuning with DFT-aggregated forces as well. Although the differences between the **Tuning** CG and standard CG with no machine learned force filed might be slight, DFT-based tuning could yield marginal improvements with respect to the primary loss function. However, it is unclear if this approach, in comparison to the original CG method, significantly enhances generalization or universal applicability, as suggested by the authors. This raises questions about the method’s novelty and practical impact, especially if it does not introduce substantial improvements over existing approaches.
2. **Limited and Insufficient Metrics**: The study's metrics are simplistic and may not support robust conclusions about the model’s generalizability or real-world applicability. For instance, the stability measure, assessed via temperature, provides only intermediate insights. A more comprehensive evaluation could follow metrics like those in [1], which evaluate atom-level stability and bond accuracy to better capture molecular coherence in 3D molecular generation. Moreover, the authors should compare their CG model to standard coarse-grained force fields, such as Martini, rather than only to their trained baseline. Additional limitations include the use of only one system each for RMSD and energy landscape evaluations, with no stable RMSD data for differentiating conformational states, where the author just quoted “Visual
inspection of the trajectory hints at the fact that the side chains of the two more distant amino acids
move further away from each other in this conformation” but not shown in the figure. Furthermore, the CG model's energy landscape lacks overlap in low-energy modes with the reference, suggesting a poor representation of molecular dynamics. A more rigorous evaluation design, as demonstrated in [2], would strengthen the validation of CG model performance.
3. **Minimal Contribution to the Machine Learning Community**: The paper’s contribution to ML is limited, as it does not address core ML topics such as optimization, representation learning, or generative modelling. Nor does it apply ML techniques innovatively to scientific problems, as seen in works like [3] and [4]. Instead, the study mainly adopts a prior CG approach with minimal innovation in priority calculation and applies the MACE model for CG force learning. This limited novelty and applicability may reduce the work’s relevance to the broader ML community.

**References:**

- [1] Hoogeboom, Emiel, et al. "Equivariant diffusion for molecule generation in 3d." *International Conference on Machine Learning.* PMLR, 2022.
- [2] Charron, Nicholas E., et al. “Navigating protein landscapes with a machine-learned transferable coarse-grained model.” *Nature Communications*, 2023.
- [3] Li, Shaoning, et al. "F $^ 3$ low: Frame-to-Frame Coarse-grained Molecular Dynamics with SE (3) Guided Flow Matching." *arXiv preprint arXiv:2405.00751,* 2024.
- [4] Arts, Marloes, et al. "Two for one: Diffusion models and force fields for coarse-grained molecular dynamics." *Journal of Chemical Theory and Computation* 19.18 (2023): 6151-6159.

**Questions:**

1. Could the authors provide more detail on how the CG parameters were tuned? Specifically, was the tuning based on MLFF-predicted forces or on DFT-aggregated atomic forces? Understanding this would help clarify the basis of the tuning choices and how they contribute to model accuracy.

2. For stability assessment, could the authors consider more comprehensive indicators beyond temperature alone? For example, studies like [1] (Hoogeboom et al., 2022) use atom-level stability and bond-type accuracy, which might provide additional insights into molecular coherence and CG model reliability.

3. Additionally, including a comparison to a well-established CG force field, such as Martini, might provide a useful reference point for assessing performance.

4. The RMSD and energy landscape analyses are insightful. Would the authors consider expanding these analyses to more than one system? Multiple systems could reinforce the stability and generalizability of the model across different molecular configurations.

5. In the energy landscape analysis, there seems to be limited overlap in low-energy modes between the CG model and the reference. Could the authors share their perspective on this outcome and any potential implications for MD applications?

6. The paper makes good strides in applying MACE to CG molecular dynamics. Could the authors further clarify how this work fits into broader ML themes, such as optimization or representation learning? Discussing this may help frame the study’s relevance to the ML community.

---

> ### Author Response · Authors · 2024-11-22
>
> Thank you for your detailed comments on our manuscript, we'll respond to each of your numbered weaknesses and questions below:
>
> **Weaknesses**
>
> 1. This is a very good question and we agree that this part of the paper is lacking clarity. We optimize our CG mapping parameters with respect to the DFT-aggregated forces, or more precisely, their average magnitude across a small sample of the dataset. As we show in section 3.1 and explain in the Methodology section, this is a very good proxy for how well the MLFF model trains on the CG data. The MLFF-predicted forces would be much less informative at that stage as they are basically random, because the model has not been trained yet at that stage. We propose to add the three sentences above for clarity to section 3.1. Furthermore, we would like to point out that the extension of the CG mapping strategy by Webb et al. is only one part of our key contribution to the field. We agree that the data presented in our paper is not sufficient to prove for all upcoming use cases that the tuned approach will have significant superiority over the standard one. However, the main contribution of our work is that we apply this CG mapping strategy to build a transferable CG-MLFF model that is not restricted to very specific systems like proteins (where the CG mapping can be hardcoded). Furthermore, we apply the MACE architecture in the CG context which has not been done previously.
>
> 2. We agree with this criticism partly in the sense that more evaluations and metrics are required to demonstrate high stability and robustness of a CG-MLFF approach. However, we believe that this is outside the scope of our proof-of-concept study presented in the manuscript. For our responses on some of the details of the review above, please see our responses below to the specific questions the reviewer posted.
>
> 3. We refer the reviewer to our response to Weaknesses for Reviewer iQM8.
>
> **Questions:**
>
> 1. This question has been addressed in the response to the weaknesses above. We agree that more clarity in the description is needed and it will be added.
>
> 2. We agree that additional stability metrics would be beneficial to validate model reliability and we have looked into many different metrics as well as the atom-level stability. However, the main issue with many of these are that they are defined for all-atom simulations, where for example, the correctness of the neighborhood of an atom can be easily determined at each time step. For CG systems, we attempted to come up with a generalization of such metrics but did not succeed in a consistent way as comparison with Martini trajectories demonstrated. Of course, we remain interested in dedicating more time into developing such evaluation scores for CG MD trajectories in the future.
>
> 3. We agree that a quality assessment of the MD trajectories by comparison to either Martini or a classical all-atom force field is of high interest, and hence, with have included such a comparison for one of our test systems, the IPP lactotripeptide, as depicted in Fig. 4 and 5. We have selected this system as it exhibits very high MD stability with our model for long simulation times (several nanoseconds) which are required for such comparisons of RMSD and TICA plots. For other systems, we have either not observed this outstanding stability or, as in the presented case of the RNA systems, observed other unphysical artifacts in the simulations. We would like to point out that we do not claim to present an application-ready superior CG model in our work, but instead present a proof-of-concept for a very novel type of model – a transferable CG-MLFF that can be applied across different (organic) chemical systems and is trained on a dataset with large chemical diversity. We see this as a significant step towards a CG-MLFF model that is usable in real-world applications. However, we also want to be transparent about the limitations of the current model which aims at generalizing across chemical systems in a way that has not yet been reported in ML-based CG modeling.
>
> 4. As transparently explained in the manuscript, there were a few more test systems we looked at in this context. Either we did obtain stable simulations, but very uninteresting sampling of conformations even for our reference methods (GFN-FF, Martini), or systems were not stable enough for long periods of simulation time. We agree that finding more interesting systems for comparison of energy landscapes would be beneficial in the long term, but we also believe that the scope of this paper was providing a proof-of-concept that generalizable CG-MLFFs are possible and for this scope we believe that the example presented in the manuscript is sufficient.
>
> 5. We would like to ask the reviewer if they could explain this observation a bit more to us. We are unsure about what this observation is referring to.
>
> 6. Please also refer to response to Weaknesses 3.

---

> ### Comment · Reviewer_ezmK · 2024-11-29
> **Official Comment by Reviewer ezmK**
>
> Apologies for the delayed response, and thank you for your follow-up. However, I will still keep my initial score.
>
> In response to the question:
>
> > *We would like to ask the reviewer if they could explain this observation a bit more to us. We are unsure about what this observation is referring to.*
>
> I recommend referring to Figures 2 and 5 in [1]. This paper provides an excellent example of how to compare two different force fields based on the landscape derived from simulated trajectories.
>
> [1] Charron, Nicholas E., et al. “Navigating protein landscapes with a machine-learned transferable coarse-grained model.” *Nature Communications*, 2023.

---

### Official Review · Reviewer_X3Ko · 2024-10-30

**Soundness:** 2
**Presentation:** 3
**Contribution:** 1
**Rating:** 3
**Confidence:** 4

**Summary:**

This work discusses a specific type of coarse graining approach for all-atom to coarse grained simulations, and uses MACE learned interatomic potentials to predict properties necessary for simulations. The key contributions are (1) the specific strategy (with some learnable parameters) for coarse-graining and (2) the use of MACE potentials rather than other ML potentials for the dynamics of the coarse-grained beads.

The primary metrics for identifying the performance of the resulting coarse-grained potential are qualitative comparisons between all-atom (ML, GFN-FF) and coarse grained (MARTINI, this work) simulations for a couple representative biomolecular systems without solvent (e.g. biomolecules in vacuum).

**Strengths:**

**Originality:** The authors use a specific coarse-graining strategy that appears to be unique in literature.

**Quality:** The authors consider a number of constraints and edge cases in how the coarse graining should be achieved, and necessary properties for the system/approach. They include some simulations of reasonably large simulations to verify the outcomes (albeit without solvent).

**Clarity:** The paper is clearly written and easy to follow in the structure and logic.

**Significance**: The process of coarse-graining (and back-mapping) are very important for the biomolecular community, but notoriously difficult to implement/improve on, and thus difficult to apply outside of very well studies areas such as water/DNA/RNA/etc. This is clearly an important research area.

**Weaknesses:**

The authors describe the key innovations as extending a cited work (Webb et al) as the specific coarse-grained mapping, and the use of MACE potentials. It is very surprising that they do not provide any qualitative or quantitative comparisons with that work to show that either the updated (tuned) strategy or the use of MACE potentials significantly improves the performance on either the original benchmarks in Webb et al, or in the case studies in this work. The only ablations here appear to be for the 'tuned' vs 'untuned' models.

A key contribution of this work is the use of MACE potentials. As the authors state on Page 2 "Furthermore, previous CG-MLFF approaches have been relying on the invariant SchNet architecture for the force field model, which for all-atom force fields has been mostly replaced by other (typically equivariant) architectures in recent studies, for example, MACE (Batatia et al., 2022b), VisNet (Wang et al., 2022), or Allegro (Musaelian et al., 2023)." However, some recent works have already considered models beyond SchNet, such as Allegro (Loose, Both et al. JPC-B 2023) or FLARE (Dushatko, Kozinsky et al. npj Comp Materials 2024). A comparison to these existing strategies should be included to show the improvement on this work over prior literature.

The accuracy of a coarse-grained potential requires more validation than a simple qualitative check compared to the all-atom potential. This is especially the case when comparing for fixed time windows, as the equivalent timescales in coarse grained potentials are typically much longer than the timestep would suggest. Thermodynamic consistency, accuracy of averaged functions like the RDF or radius of gyration or other observables would be a much better check. Also see the Opinion (Durumeric, Clementi et al, Opinion in Structural Biology, 2023) for a nice overview of the important considerations and specifically Figure 3 which highlights quantitative agreement for an all-atom and coarse grained potential.

Reproducibility: The code for the coarse-graining implementation does not appear to be available (or if it is already present in MACE-JAX, that is not clear from the text). Thus it might be difficult to reproduce or understand the details of the implementation.

**Questions:**

1. Quantitative agreement and thermodynamic consistency should be shown for reference systems in prior work. Qualitative comparisons of energy or RMSD for a fixed time window is not sufficient validation of a coarse grained simulation.

2. Ablations are needed to show the improvements of MACE vs SchNet and original (Webb et al) vs the CG implementation here would be very helpful. Alternatively, quantitative comparisons between this work and prior work would also establish the improvements in this work.

3. The authors should clarify the novelty of their work compared to prior work using potentials like Allegro (Voth et al, see citation above) or FLARE (Kozinsky et al, see citation above) and show quantitative comparisons with these prior works. This is especially the case since those prior works demonstrated quantitative agreement between all-atom and coarse grained observables for the simulations of interest.

---

> ### Author Response · Authors · 2024-11-22
>
> Thank you for your detailed comments on our manuscript, we'll respond to each of your numbered weaknesses and questions below:
>
> **Weaknesses:**
>
> **The authors should clarify the novelty of their work compared to prior work using potentials like Allegro (Voth et al, see citation above) or FLARE (Kozinsky et al, see citation above) and show quantitative comparisons with these prior works. This is especially the case since those prior works demonstrated quantitative agreement between all-atom and coarse grained observables for the simulations of interest.**
>
> The main goal of this paper was to show what a general strategy for coarse-graining, the most state-of-the-art architecture for force field prediction and a new dataset generation method allowing to include larger fragments, could lead to in terms of a general strategy to perform coarse-grained MD.
>
> It is one of the few (if not the only) paper that explores the training and generalizability of a ML-based coarse-grained force field for a wide range of chemistry (RNA, lipid, proteins) and for a wide range of molecule sizes. Furthermore, the paper doesn’t just validate the force field on a validation set, it tests the force field’s usefulness by applying it in MD simulations. Papers that do run MD simulations for validation, often just run on systems they have been specifically trained on instead of showing generalization to new systems.
> To the best of our knowledge, Allegro has neither been employed in the CG context nor in the context of being a generalizable force field yet. However, we agree that investigating this architecture in future work is an interesting direction. Regarding FLARE, we have a similar view.
>
> We agree that an evaluation on the original molecules from Webb et al. comparing the original CG scheme to our tuned one could be helpful in addition to our training comparison. However, as the molecules tested in Webb et al. are not of the same type as the ones we trained on in our work and since they trained specifically on these individual systems instead of in a transferable fashion, a comparison of the CG methods would be difficult unless the same force field would be employed.
>
> Regarding prior CG work with MD: except for coarse-grained chignolin, data are not available or deal with material science, which is outside of the ability of our model. For chignolin, our simulation was not stable enough to do a direct comparison. That being said, we will continue to search for more reasonable comparisons in future work as we agree that these are strongly required to assess a CG force field model. Moreover, in most prior work, specific force fields have been trained for those specific molecules, which we reemphasize makes the comparison difficult.
>
> We should definitely add more all-atom vs. CG comparisons in the future, possibly using MACE-OFF for small-enough molecules in our test set and classical force field for larger molecules. However, we emphasize that we explicitly added a comparison of our tripeptide test system with the all-atom force field GFN-FF. We also did a comparison of this test case with MARTINI, but we also agree that more such comparisons must follow in the future, but were not possible within the scope of our current model. We also thank the reviewer for their useful paper recommendations.
>
> **Questions**
>
> **2. Ablations are needed to show the improvements of MACE vs SchNet and original (Webb et al) vs the CG implementation here would be very helpful. Alternatively, quantitative comparisons between this work and prior work would also establish the improvements in this work.**
>
> We consider the comparison of performance of different force field architectures in a CG context to be out of scope for this paper. As explained above, quantitative comparison to prior work is difficult but additional all-atom vs. CG comparisons and CG-MLFF vs. MARTINI can be explored.
>
> **1. Quantitative agreement and thermodynamic consistency should be shown for reference systems in prior work. Qualitative comparisons of energy or RMSD for a fixed time window is not sufficient validation of a coarse grained simulation.**
>
> Same as above.

---

> > ### Comment · Reviewer_X3Ko · 2024-11-22
> > **Relevant papers**
> >
> > I cited two papers in my review of previous works that have used ML architectures for coarse-grained structures.
> > * (DeepMD/Allegro) Loose et al. https://doi.org/10.1021/acs.jpcb.3c05928
> > * (FLARE) https://doi.org/10.1038/s41524-023-01183-5
> >
> > These were found with a short literature search; I probably missed other examples in the literature of others combining various machine learning potentials with/for/on coarse-grained structures, for example (among a number of papers from various combinations of Durumeric, Clementi, Noe):
> > * Durumeric et al. https://doi.org/10.48550/arXiv.2407.01286

---

> > > ### Author Response · Authors · 2024-11-25
> > >
> > > We thanks the reviewer for those 3 references, and realize that we misunderstood the reviewer's comment.
> > >
> > > Allegro has indeed been applied to learn a force field over the center of mass of water molecules, which is a very specific type of coarse-graining. This should be part of our literature review, but as it differs a lot from what we are trying to accomplish, we believe that a direct comparison with our work is not sensible. We will add this reference to our manuscript.
> > >
> > > For the other papers, we will also add them to our literature review, but please note that we are already acknowledging the use of CGSchnet. On this particular paper, we will also check how their two test proteins behave under our model. Finally, the uncertainty-driven active learning of CG free energy models uses GP for their force field architecture, which is an interesting development that should be mentioned in our paper. However, the test molecules presented do not suit our model abilities, and we would have to run their model on our test set, which is not feasible given out time constraints.
> > >
> > > To summarize, all those references focus on the ML-based force fields but don’t touch the CG part, and a proper benchmarking of those models directly applied to our tests or applied to our test under our CG rules, would be a completely new paper. These papers are usually very fine-tuned towards a certain type of molecules with a very clearly pre-defined (hardcoded) CG mapping where our main innovation is training a transferable CG-MLFF on a universally applicable CG algorithm (at least, for bio-organic systems). Furthermore, we are the first to apply MACE to CG, and it is in this light, that we probably misunderstood the reviewer's comment.

---

### Official Review · Reviewer_iQM8 · 2024-11-04

**Soundness:** 2
**Presentation:** 2
**Contribution:** 2
**Rating:** 5
**Confidence:** 2

**Summary:**

The paper proposes a new CG mapping method which builds over[1]. The authors demonstrate that their CG mapping outperforms standard mapping approaches by evaluating it with the MACE force field [2]. To substantiate the performance gains, they generate their own dataset, showcasing the improvements achieved by their CG mapping compared to the standard method.

references.
[1]https://www.osti.gov/servlets/purl/1558004
[2]https://github.com/ACEsuit/mace-jax

**Strengths:**

1. Coarse-graining is a critical challenge, as efficiently reducing the complexity of large bimolecular systems is essential for feasible modelling.

2. I commend the authors for identifying the lack of suitable datasets and generating their own to address this gap.

**Weaknesses:**

1. The paper claims{line 74 to 80}:
                      a) Generalizibility across chemically different molecules -> Which result in paper supports this claim?
                      b) Transferable across system size  -> Isn't this property of coarse graining itself? What unique contribution their approach makes beyond the inherent properties of coarse-graining methods?

2. Its not clear to me what exactly is the core contribution of this paper in terms of proposing a new method. What is different from proposed in[1]. Can you please clarify this using a pseudo code(Highlighting the lines which is original to this work)? For instance - provide a side-by-side comparison or annotated pseudocode highlighting the key differences and innovations compared to the reference work [1]

3. I see this paper has some development components but mainly an application paper. The selection of MACE as the modelling framework is a good choice, given its status as a top-performing model [2]. Adding more force fields and datasets on evaluation will considerably strengthen the paper. I suggest using more equivariant potentials like allegro[3] and nequip[4]. Additional types of dataset may include presence of diverse atoms/bonds in the molecule.

references:
[1] https://www.osti.gov/servlets/purl/1558004
[2] https://matbench-discovery.materialsproject.org/
[3]https://www.nature.com/articles/s41467-023-36329-y
[4]https://github.com/mir-group/nequip

**Questions:**

Please see weaknesses as well.

1. Figure 3. shows the training loss curve and  tuned model is doing well. Do you have the test loss so we know it is not just overfitting?

2. For figure 5. can you also please include comparison to standard model. Also it will appreciable if the performance can be quantified.

---

> ### Author Response · Authors · 2024-11-22
>
> Thank you for your detailed comments on our manuscript, we'll respond to each of your numbered weaknesses and questions below:
>
> **Weaknesses:**
>
> 1. In the field of machine learning force fields (MLFFs), we generally distinguish between system-focused and transferable MLFF approaches. In the former, the model is just trained on a single molecule in different conformations (a single potential energy surface) with the goal to later obtain accurate simulations for this molecule. On the contrary, transferable approaches train on a dataset with a variety of systems with the goal to be applicable to unseen molecules of similar chemistry.
> In the field of CG-MLFFs, the large majority of published work has been done in the system-focused setup which is common for proof-of-concept studies. One of the reasons for this is that a generally applicable CG mapping is not trivial to define, however, as a result we are lacking understanding of whether a truly transferable CG-MLFF approach is even feasible. Due to the difficulty of defining a general CG mapping, the few published transferable CG-MLFF approaches are restricted to very specific types of systems, typically proteins where the CG mapping can be fixed for each amino acid in a simple rule-based way.
> Our presented approach is novel in the aforementioned sense. Our CG mapping strategy works for any (organic) chemical system, which is reflected not only in the fact that our dataset contains lipids, RNA and proteins, but also that due to the fragmentation-based dataset generation we train on a diverse set of organic fragments (including molecules from the PubChem database). This is the generalizability that we claim at the end of our Introduction.
> Transferability across system sizes is another important topic for MLFFs. The ability to train on system sizes in a the range of 20 to 200 atoms per system, but applying the learnt model to systems of 1000s of atoms. This is an MLFF-related challenge and not related to the coarse-graining problem. For all-atom force fields, this is already a difficult task and for CG-MLFF, to the best of our knowledge, it has never been shown possible. We show in our paper that our model is able to do this by running an MD on the 2Z75 system of >3000 atoms – however, we acknowledge that a detailed analysis of robustness across a large number of such systems was beyond the scope of our work.
> We propose to add a paragraph to highlight our core contribution more prominently in our introduction
>
> 2. Our paper is both (i) applying the CG mapping strategy of [1] in a new context that has not been done before (transferable MACE-based CG-MLFFs), and (ii) extending the method proposed in [1] by adding tunability. It is important to highlight that the combination of these two aspects is our key contribution to the applied ML community. First and foremost, we present the first transferable CG-MLFF approach that is not limited to a single type of system like proteins for which the CG mappings can be hardcoded. This feature is enabled by the versatile CG mapping strategy proposed in [1]. Furthermore, we add two additional contributions, (a) the extension of [1] by adding the tunability of the spectral decomposition (multiple Laplacians) and (b) using MACE as the force field architecture which has not been applied to the CG use case before. We propose to highlight our core contribution more prominently in our introduction and abstract.
>
> 3. As outlined above, our approach contains several new features compared to previous CG-MLFF work. As a result, we decided to concentrate on specific aspects of the pipeline for ablation studies, most prominently, comparing the standard CG mapping strategy close to the one reported in [1] with our tunable extension. Adding a comparison of MACE with other equivariant potentials like the ones you mentioned are definitely also interesting but were not in our focus for this work. Unfortunately, as model training takes multiple weeks on our large dataset, we are not able to add such comparisons to the paper at this moment but would be interested in revisiting this aspect in future work. We also propose to add a sentence explaining this to the paper in section 2.1.
>
> **Questions**
>
> 1. The presented curve does not show the data collected on the training set but rather for the validation set. This is mentioned in the caption, but we assume the word “Training curves” has led to confusion. We would therefore like to rephrase it to “Validation set metrics for the standard and tuned model during training.”.
>
> 2. The same plot for the standard model is shown in Figure 8 in Appendix A.7 due to the space constraints in the paper. We propose to change the paper to mention this not only in the text but also in the figure caption. This should make it easier to find. We are not aware of any straightforward way to quantify the agreement of TICA plots that has been applied in the literature before. Are you aware of any metric that we could compute?

---

> > ### Comment · Reviewer_iQM8 · 2024-11-26
> >
> > Thank you for answering my questions. I will keep the score at 5 as I still think the novelty of work is not at par with ICLR standards.

---

### Meta-Review · Area_Chair_GH9X · 2024-12-22

**Metareview:**

In this work, authors present a transferable deep learning approach for coarse-graining (CG) molecular dynamics simulations of biological systems. The key contributions include: (1) extending an existing graph-based CG mapping method with tunable parameters, (2) applying the MACE architecture for force field prediction in the CG context, and (3) developing a new dataset generation approach using molecular fragmentation. The authors demonstrate their method on systems including proteins, RNA, and lipids.

The research addresses an important challenge in molecular simulation, that is, developing transferable CG models that work across different types of biomolecules. The strengths include a novel combination of modern ML architectures (MACE) with CG approaches, creation of a diverse training dataset, and some promising initial results showing qualitative agreement with reference simulations for certain test cases.

The work has several weaknesses as follows. First, as noted by Reviewer X3Ko, the paper lacks sufficient validation and comparison with existing methods. The authors acknowledge they have not compared their approach quantitatively with prior CG methods or demonstrated thermodynamic consistency. While they explain this is challenging due to the broader scope of their method, robust validation is essential for establishing the value of a new CG approach. Second, as highlighted by Reviewer ezmK, the metrics and evaluation are limited. The stability assessment relies mainly on temperature, with minimal quantitative analysis of conformational sampling or comparison to reference methods like MARTINI. The energy landscape analysis shows concerning lack of overlap in low-energy modes with reference calculations. Third, both Reviewers iQM8 and ZwoN question the degree of technical novelty beyond prior work. While the authors argue their key contribution is demonstrating transferability across different biomolecule types, the improvements from their tunable CG mapping extension are not clearly established. The application of MACE, while novel for CG, is not extensively validated against other architectures. The authors provided detailed responses acknowledging many of these limitations while arguing their work represents an important proof-of-concept toward transferable CG models.

However, for acceptance at ICLR, the paper needs stronger technical innovation in the ML methodology and more rigorous validation demonstrating clear advantages over existing approaches. The current work appears more suitable for a domain-specific journal focusing on molecular simulation methods.

**Additional Comments On Reviewer Discussion:**

Reviewer iQM8 questioned the core novelty and generalizability claims, particularly regarding transferability across chemically different molecules and system sizes. The authors clarified that their contribution lies in developing the first transferable DL-based CG force field approach that works across diverse biosystems, rather than being limited to specific system types. However, iQM8 maintained their score of 5, citing insufficient novelty for ICLR standards.

Reviewer X3Ko raised concerns about lack of comparisons with prior work (Webb et al.) and other ML architectures like Allegro and FLARE. The authors explained that direct comparisons were challenging due to differences in training approaches and system types, noting that their focus was on demonstrating general applicability across diverse chemistry rather than optimizing for specific systems.

Reviewer ezmK provided detailed critique of the evaluation metrics and parameter tuning approach. The authors clarified that they optimize CG parameters using DFT-aggregated forces rather than MLFF-predicted forces, and acknowledged the need for more comprehensive stability metrics while explaining the challenges in developing such metrics for CG systems.

Reviewer ZwoN raised technical questions about the method's implementation and stability. The authors provided detailed explanations of their approach's theoretical foundations and addressed concerns about potential instability in simulations.

These discussions informed my rejection decision by highlighting that while the authors could explain their design choices and limitations, they were unable to address the core concerns about insufficient validation and comparative evaluation.

---

### Decision · Program_Chairs · 2025-01-22

Reject